# BPG4 regulates chloroplast development and homeostasis by suppressing GLK transcription factors and involving light and brassinosteroid signaling

Ryo Tachibana [1], Susumu Abe[2,3], Momo Marugami[2,3], Ayumi Yamagami[1], Rino Akema[1], Takao Ohashi[1], Kaisei Nishida[1], Shohei Nosaki [4], Takuya Miyakawa [1], Masaru Tanokura [5], Jong-Myong Kim[2,5,6], Motoaki Seki [2], Takehito Inaba[7], Minami Matsui [2], Kentaro Ifuku [8], Tetsuo Kushiro[3], Tadao Asami [5] & Takeshi Nakano [1,2] ✉

Chloroplast development adapts to the environment for performing suitable photosynthesis. Brassinosteroids (BRs), plant steroid hormones, have crucial effects on not only plant growth but also chloroplast development. However, the detailed molecular mechanisms of BR signaling in chloroplast development remain unclear. Here, we identify a regulator of chloroplast development, BPG4, involved in light and BR signaling. BPG4 interacts with GOLDEN2-LIKE (GLK) transcription factors that promote the expression of photosynthesis-associated nuclear genes (*PhANGs*), and suppresses their activities, thereby causing a decrease in the amounts of chlorophylls and the size of light-harvesting complexes. *BPG4* expression is induced by BR deficiency and light, and is regulated by the circadian rhythm. *BPG4* deficiency causes increased reactive oxygen species (ROS) generation and damage to photosynthetic activity under excessive high-light conditions. Our findings suggest that BPG4 acts as a chloroplast homeostasis factor by fine-tuning the expression of *PhANGs*, optimizing chloroplast development, and avoiding ROS generation.

Chloroplasts, organelles unique to plant cells, have important roles in performing photosynthesis, a vital process for plant growth. Through photosynthesis, plants contribute to carbon fixation, oxygen production, and thereby maintenance of the environmental condition on the earth. Photosynthesis activity is tightly linked with chloroplast development, and the chloroplasts in plant cells are exposed to dynamic changes of their surroundings. Therefore, plants finely regulate chloroplast development in response to environmental cues for optimizing photosynthetic efficiency. Chloroplast biogenesis and development are determined by products of both photosynthesis-associated plastid genes (*PhAPGs*) and photosynthesis-associated nuclear genes (*PhANGs*). Several essential factors for photosynthesis

[1]Graduate School of Biostudies, Kyoto University, Kitashirakawa-Oiwake-cho, Sakyo-ku, Kyoto 606-8502, Japan. [2]CSRS, RIKEN, Tsurumi-ku, Yokohama 230-0045, Japan. [3] School of Agriculture, Meiji University, Tama-ku, Kawasaki 214-8571, Japan. [4]Faculty of Life and Environmental Sciences, Tsukuba University, 1-1-1 Tennoudai, Tsukuba-shi 305-8572, Japan. [5]Graduate School of Agricultural and Life Sciences, University of Tokyo, Bunkyo-ku, Tokyo 113-8657, Japan. [6]Ac-Planta Inc., Bunkyo-ku, Tokyo 113-0044, Japan. [7]Department of Agricultural and Environmental Sciences, Faculty of Agriculture, University of Miyazaki, 1-1 Gakuenkibanadai-nishi, Miyazaki 889-2192, Japan. [8]Graduate School of Agriculture, Kyoto University, Sakyo-ku, Kyoto 606-8502, Japan. ✉e-mail: nakano.takeshi.6x@kyoto-u.ac.jp

such as *Photosystem II D1 protein* (*psbA*) and *Ribulose 1,5-bisphosphate carboxylase/oxygenase large subunit* (*rbcL*) are encoded in the chloroplast genome as *PhAPGs*, and their expression is regulated by chloroplast-localized factors such as sigma factors and CND41[1,2]. In contrast, the majority of crucial factors determining chloroplast development such as chlorophyll biosynthesis-related enzymes, and subunits of light-harvesting complex (LHC) and photosystem I/II (PSI/II) are encoded in the nuclear genome as *PhANGs*[3]. Transcription factors including GOLDEN2-LIKE 1/2 (GLK1/2), GATA NITRATE-INDUCIBLE CARBON-METABOLISM-INVOLVED (GNC), and CYTOKININ-RESPONSIVE GATA1 (CGA1)/GNC-LIKE (GNL) have been recognized as key regulators of *PhANG* expression[3-6]. GLK transcription factors belong to the GARP family (Golden2, ARR-B and PSR1), and have been investigated in various species, such as *Arabidopsis thaliana* (hereafter Arabidopsis), tomato, rice, maize, barley, moso bamboo, and *Physcomitrium patens*[7-12]. In Arabidopsis, GLK1 and GLK2 function redundantly, and play a critical role in normal chloroplast development[6,13]. GLKs directly bind to the promoter regions of *PhANGs*, thereby promoting chlorophyll biosynthesis, the assembly of the photosynthetic apparatus, and subsequently, chloroplast development[6]. GLKs have been reported to be associated with light signaling, retrograde signaling, and several phytohormone signaling, including auxin, cytokinin, and brassinosteroids (BRs)[14-20].

BRs have various physiological activities, such as promoting cell division and elongation, increasing abiotic stress resistance and disease resistance, and regulating chloroplast development[21,22]. In comparison with wild-type (WT) plants, BR-deficient mutants such as *de-etiolated 2* (*det2*) grown in the dark exhibit de-etiolated phenotypes with shortened hypocotyls; open cotyledons; and increased expression of photosynthesis-associated genes such as *Ribulose 1,5-bisphosphate carboxylase/oxygenase small subunit* (*RbcS*), *Light-harvesting complex protein/chlorophyll a/b binding protein* (*LHC/CAB*), and *psbA*[23]. In the light, *det2* mutants produce dwarf, dark green leaves and exhibit promotion of chloroplast development[20,23]. The phenotypes of the BR receptor-deficient mutant *brassinosteroid-insensitive 1* (*bri1*) and the BR signaling negative kinase BIN2 gain-of-function mutant *brassinosteroid-insensitive 2-1* (*bin2-1*) are similar to that of *det2*[24,25]. Based on these phenotypes, BR has significant functions in chloroplast development, but knowledge of direct and key mechanisms to regulate chloroplast development under BR signaling is limited.

To clarify the unknown mechanisms underlying BR signaling, we used Brz as a specific inhibitor of BR biosynthesis, the compound which we synthesized. Brz specifically inhibits DWARF4, a cytochrome P450 monooxygenase in the BR biosynthesis pathway, by the affinity of its triazole ring for the heme position of cytochrome P450 monooxygenase[26,27]. Brz treatment causes a decrease in endogenous BR contents and dwarf phenotypes similar to those of *det2* and *bri1*[26]. WT plants germinated in the dark have elongated hypocotyls and closed cotyledons. In contrast, WT plants treated with Brz in the dark have shortened hypocotyls and opened cotyledons, which resembles photomorphogenesis. By exploiting the effects of Brz on hypocotyls in the dark, we screened several *Brz-insensitive-long hypocotyl* (*bil*) mutants, which had significantly longer hypocotyls than the WT plants germinated on medium containing Brz in the dark[28-31]. *BRZ-INSENSITIVE-LONG HYPOCOTYL 1* (*BIL1*)/*BRASSINAZOLE-RESISTANT 1* (*BZR1*) was identified through the screening of *bil* mutants, and subsequent studies revealed the molecular function of BIL1/BZR1 to be a master transcription factor that regulates plant development via BR signaling[28,32,33].

In the light, dark green and dwarf-types leaves and promotion of chloroplast development were observed in the WT seedlings grown on medium supplemented with Brz[34]. By exploiting the effects of Brz on chloroplasts in the light, we screened *Brz-insensitive-pale green* (*bpg*) mutants, whose green leaves were paler in color than those of WT plants grown with Brz in the light[35-37]. BPG1, BPG2, and BPG3 were

identified from these *bpg* mutants and have been reported to be regulators of chloroplast function downstream of BR signaling[35-37]. *BPG1* is encoded by the same gene responsible for *dvr* mutants, and BPG1/DVR is a chlorophyll biosynthesis-related enzyme, 3, 8-divinyl protochlorophyllide *a* 8-vinyl reductase[35,38]. BPG2 is a chloroplastic GTPase that is evolutionarily conserved in photosynthetic organisms and bacteria. BPG2 plays an essential role in chloroplast rRNA maturation[36]. BPG3 is a chloroplast protein that is conserved in higher plants, algae, and photosynthetic bacteria and has a crucial effect on the photosynthetic activity of PSII[37]. The expression of *BPG1*, *BPG2*, and *BPG3* is induced in response to BR deficiency caused by Brz treatment, implying that their expression is regulated by BR signaling[35-37]. These investigations suggest that BPG1, BPG2, and BPG3 play important roles in the regulatory mechanism of chloroplast development downstream of BR signaling.

In this report, we identified a regulator of chloroplast development, BPG4, from a newly discovered *bpg* mutant, which regulated GLK transcriptional activity and involved light and BR signaling. Detailed analysis of BPG4 revealed the unknown molecular mechanisms that regulate chloroplast development and homeostasis to perform suitable photosynthesis.

## Results

### Isolation of *bpg4-1D* mutants

Brz is a specific inhibitor of BR biosynthesis that decreases the internal BR content in plants. Plants grown on medium supplemented with Brz exhibit de-etiolated phenotypes in the dark and increased expression of photosynthesis-associated genes such as the chlorophyll biosynthesis-related genes, *LHC*, *rbcS*, and *psbA*, and when those plants are grown in the light, they produce dark green leaves[35,36]. Despite the obvious physiological effects of BRs on chloroplast development, the molecular mechanism of how BR signaling regulates chloroplast development has not yet been elucidated. To determine the unknown mechanisms, we used Brz to screen approximately 10,000 Arabidopsis Full-length cDNA Over-eXpressing (FOX) transgenic lines[39] and isolated a gain-of-function mutant, *Brz-insensitive-pale green 4-1D* (*bpg4-1D*). Cotyledons of the WT plants grown with Brz were darker green, which resulted from increased endogenous chlorophyll contents, than those in normal conditions without Brz (Fig. 1a, b). The cotyledons of the *bpg4-1D* mutants were paler green in color than those of the WT plants grown with Brz in the light. The endogenous chlorophyll contents of *bpg4-1D* were approximately 60% of those of the WT plants and were not increased by Brz treatment (Fig. 1a, b). These results suggest that *BPG4*, a causal gene of *bpg4-1D*, seems to have important functions in chloroplast regulation by BR signaling.

FOX lines are transgenic lines in which Arabidopsis full-length cDNA was expressed under the control of the cauliflower mosaic virus 35S (*CaMV 35S*) promoter and nopaline synthetase (NOS) terminator[39]. To identify the causal gene of the *bpg4-1D* mutant, we determined the cDNA inserted in *bpg4-1D* via PCR with specific primers for the *35S* promoter and NOS terminator; the results revealed that cDNA encoding *At3g55240* was inserted in *bpg4-1D*. Significant overexpression of *At3g55240* in *bpg4-1D* compared with that in the WT plants was determined by qRT–PCR (Fig. 1c). To confirm that *At3g55240* overexpression is responsible for the pale green phenotype of *bpg4-1D*, transgenic plants overexpressing the *At3g55240* full-length coding DNA sequence (CDS) under the *35S* promoter were generated. When treated both with Brz and without Brz, the *At3g55240*-overexpressing plants exhibited pale green phenotypes similar to those of the *bpg4-1D* mutants (Supplementary Fig. 1a, b). Therefore, we named *At3g55240 BPG4*.

### *BPG4* encodes a functionally unknown protein that is evolutionally conserved across land plants

In the latest publications, *BPG4* has been reported as *Pseudo-Etiolation in Light* (*PEL*) and *REPRESSOR OF PHOTOSYNTHETIC GENES 2*

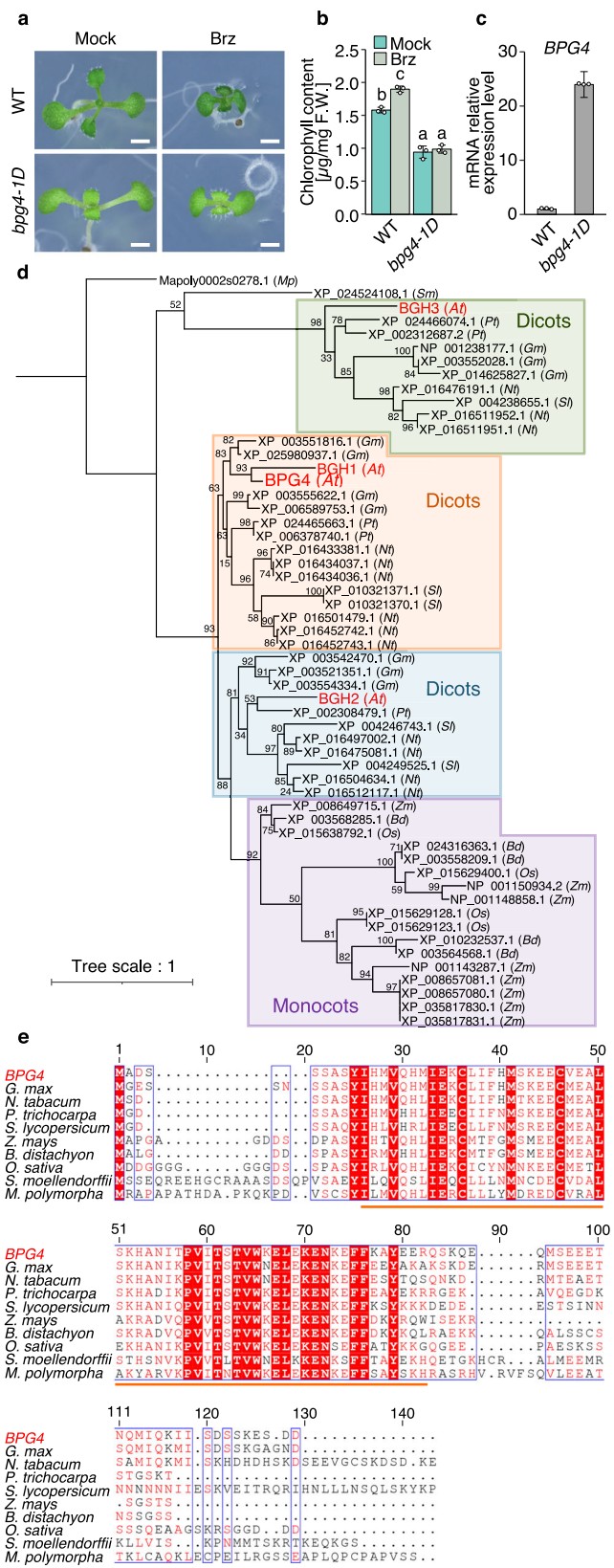

**Fig. 1 | Identification and molecular characterization of *BPG4*. a** WT (Col-0) and *bpg4-1D* seedlings grown on half-strength MS medium supplemented with 1 µM Brz or the same volume of dimethyl sulfoxide (DMSO) solvent (mock) for 9 days. Scale bars = 1 mm. **b** Endogenous contents of the total chlorophyll of WT and *bpg4-1D* plants grown in the presence of 1 µM Brz or the same volume of DMSO solvent (mock) for 14 days. The means and standard deviations (SDs) were obtained from 100 mg plant tissue, and three biological replicates. The different letters above the bars indicate statistically significant differences between the samples (two-way ANOVA: Tukey–Kramer's multiple comparisons test, *P* < 0.001), and groups with the same letters are not significantly different. **c** qRT–PCR analysis results of *BPG4* expression in WT and *bpg4-1D* grown on soil for 7 weeks. The relative expression levels were normalized to *ACTIN2* (*ACT2*). Data are presented as the means ± SDs of three technical replicates. **d** Phylogenetic tree of *Arabidopsis thaliana* BPG4 (At3g55240), BGH1 (At3g28990), BGH2 (At5g02580), and BGH3 (At1g10657) and related proteins in other species (*Gm, Glycine max; Pt, Populus trichocarpa; Sl, Solanum lycopersicum; Nt, Nicotiana tabacum; Bd, Brachypodium distachyon; Zm, Zea mays; Os, Oryza sativa; Sm, Selaginella moellendorffii; Mp, Marchantia polymorpha*). **e** Sequence alignment of *Arabidopsis thaliana* BPG4 and BPG4 homologous genes in other plant species. The bars with orange under the sequence indicate the A_thal_3526 domain. Accession numbers: *BPG4*, NP_191084.1; *G. max*, XP_025980937.1; *N. tabacum*, XP_016434036.1; *P. trichocarpa*, XP_002308479.1; *S. lycopersicum*, XP_004249525.1; *Z. mays*, NP_001148858.1; *B. distachyon*, XP_003558209.1; *O. sativa*, XP_015629123.1; *S. moellendorffii*, XP_024524108.1; *M. polymorpha*, Mapoly0002s0278.1.

function downstream of PHYTOCHROME-INTERACTING FACTOR (PIF)[41–43], but the detailed molecular function of BGH2/RPGE1 has not been revealed. *BPG4* homologous genes were not identified in animals, nonphotosynthetic bacteria, photosynthetic *Cyanobacteria*, the algae *Chlamydomonas reinhardtii* and *Chlorella vulgaris* but were identified in various kinds of land plants, such as dicots (*Nicotiana tabacum, Solanum lycopersicum, Glycine max*, and *Populus trichocarpa*), monocots (*Oryza sativa, Brachypodium distachyon*, and *Zea mays*), and nonangiosperms (*Marchantia polymorpha* and *Selaginella moellendorffii*) (Fig. 1d, e). A phylogenetic tree of *BPG4* was constructed, which comprised approximately three major clades (Fig. 1d). The *BPG4* and *BGH1* clade contained only dicots, while the *BGH2* clade contained both dicots and monocots. The *BGH3* clade contained only dicots but was most evolutionarily close to ferns and liverworts in three clades. The alignment of BPG4 with BPG4 homologous genes in various plant species suggested that BPG4 has an evolutionarily conserved domain, which was identified as A_thal_3526 (https://www.ebi.ac.uk/interpro/entry/pfam/PF09713/) (Fig. 1e), and no other functional domains were found. These results suggested that *BPG4* is widely conserved across land plants.

## *BPG4* deficiency increased chlorophyll contents and promoted chloroplast development

To analyze the physiological function of BPG4, we isolated the knockout mutants *bpg4-1* (CS927130) and *bpg4-2* (CS391583) from a T-DNA insertion mutant pool of the Arabidopsis Biological Resource Center (ABRC) and used *BPG4*-overexpressing plants (*BPG4-OX-2* and *BPG4-OX-10*) described in the previous section (Fig. 2a, Supplementary Fig. 2a). To confirm the *BPG4* expression level in *BPG4*-overexpressing plants (*BPG4-OX-2* and *BPG4-OX-10*) and *BPG4* knockout plants (*bpg4-1* and *bpg4-2*), qRT–PCR and immunoblot analysis using an anti-BPG4 antibody were performed. Excessive amounts of *BPG4* mRNA and BPG4 protein in *BPG4-OX-2* and *BPG4-OX-10* and their decreased levels in *bpg4-1* and *bpg4-2* compared with the WT plants were detected (Fig. 2b, c). Compared with those of the WT seedlings, the cotyledons of the *BPG4-OX-2* and *BPG4-OX-10* seedlings were paler green, while those of *bpg4-1* and *bpg4-2* were darker green (Fig. 2a). At the mature growth stage, *BPG4-OX-2* and *BPG4-OX-10* also possessed paler green rosette leaves than those of the WT, and *bpg4-1* and *bpg4-2* produced darker green leaves (Supplementary Fig. 3a, b). Additionally, *bpg4-1* and *bpg4-2* exhibited late-flowering phenotypes based on rosette leaf

(*RPGE2*)[39,40], but its detailed molecular function has not been investigated. BLAST searches based on the BPG4 amino acid sequence revealed three similar genes (*At3g28990, At5g02580*, and *At1g10657*) in Arabidopsis, and we named them *BPG4 Homologous gene 1, BPG4 Homologous gene 2*, and *BPG4 Homologous gene 3* (*BGH1, BGH2*, and *BGH3*, respectively) (Fig. 1d). *BGH2* has been reported as *RPGE1* to

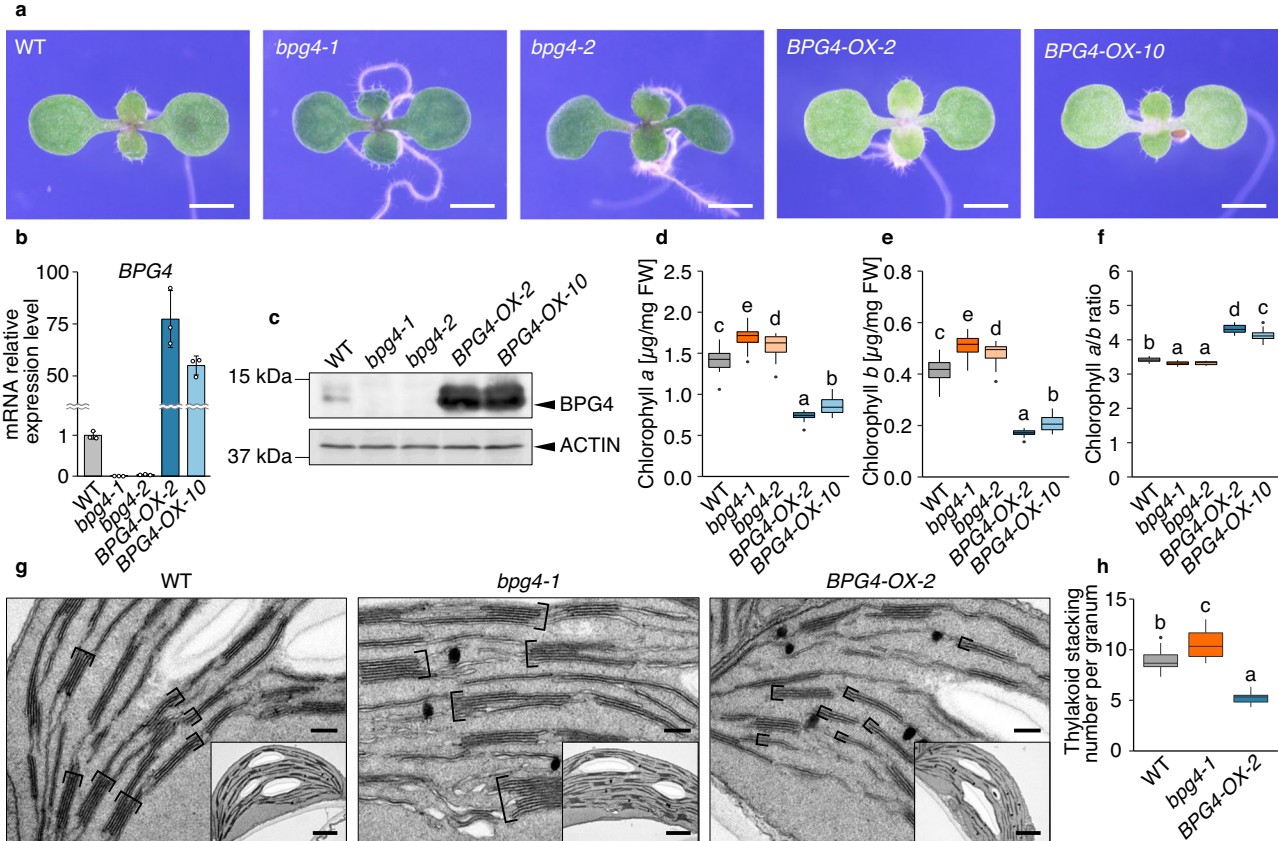

**Fig. 2 | *BPG4*-overexpressing plants showed the suppression of chloroplast development and chlorophyll contents which, conversely, were increased in *BPG4* knockout mutants. a** WT (Col-0), *bpg4-1*, *bpg4-2*, *BPG4-OX-2*, and *BPG4-OX-10* seedlings grown on half-strength MS medium for 8 days. Scale bars = 1 mm. **b** Relative expression of *BPG4* in WT, *bpg4-1*, *bpg4-2*, *BPG4-OX-2*, and *BPG4-OX-10* grown on half-strength MS medium for 8 days. The relative expression levels were normalized to *GLYCERALDEHYDE 3-PHOSPHATE DEHYDROGENASE* (*GAPDH*). The means and SDs were obtained from three biological replicates. **c** Detection of BPG4 proteins in WT, *bpg4-1*, *bpg4-2*, *BPG4-OX-2*, and *BPG4-OX-10* grown on half-strength MS medium under continuous light for 7 days. ACTIN was used as a loading control. Endogenous contents of chlorophyll *a* (**d**) and chlorophyll *b* (**e**) and the chlorophyll *a/b* ratio (**f**) in WT, *bpg4-1*, *bpg4-2*, *BPG4-OX-2*, and *BPG4-OX-10* seedlings grown on half-strength MS medium for 10 days; *n* = 20 biologically independent samples. **g** Electron microscopy image of WT, *bpg4-1*, and *BPG4-OX-2* chloroplasts in mesophyll cells of rosette leaves. The plants were grown on soil for 3 weeks. Scale bars = 0.2 μm and 1 μm (inset). **h** Quantification of grana stacking in mesophyll cells from the genotypes shown in (**g**). A thylakoid is defined as a pair of membranes enclosing a single luminal space. The means and SDs were obtained from 15 chloroplasts. Thylakoid stacking numbers were counted within the 1st, 2nd, and 3rd largest granum each in a chloroplast, and the averages obtained for the three grana were used for analysis. In (**d**–**f**), and h, the center of the boxplot is denoted by the median, a horizontal line dividing the box into two equal halves. The bounds of the box are defined by the lower quartile (25th percentile) and the upper quartile (75th percentile). The whiskers extend from the box and represent the data points that fall within 1.5 times the interquartile range (IQR) from the lower and upper quartiles. Any data point outside this range is considered an outlier and plotted individually. The different letters above the bars indicate statistically significant differences between the samples (one-way ANOVA: Tukey–Kramer's multiple comparisons test, *P* < 0.01).

number and days to bolting, while *BPG4-OX-2* and *BPG4-OX-10* exhibited early-flowering phenotypes (Supplementary Fig. 3b, d, e). In the final height of inflorescences, obvious differences were not observed in WT, *BPG4* knockout and overexpressing plants (Supplementary Fig. 3c). For a detailed analysis of the leaf color phenotypes, the endogenous contents of chlorophyll *a* and *b* were measured in WT plants, *BPG4*-knockout plants, and *BPG4*-overexpressing plants (Fig. 2d, e). The endogenous contents of chlorophyll *a* and *b* in *bpg4-1* and *bpg4-2* increased compared with those in the WT plants, while those in *BPG4-OX-2* and *BPG4-OX-10* decreased (Fig. 2d, e). These results suggest that BPG4 functions to decrease the endogenous chlorophyll contents in plants.

Furthermore, to reveal the role of BPG4 in chloroplast development, we calculated the chlorophyll *a/b* ratio and observed the chloroplast structure in *BPG4* knockout and overexpression plants (Fig. 2f–h). Generally, the chlorophyll *a/b* ratio indicates the size of LHC because chlorophyll *b* is located only in the LHC; this is in contrast to chlorophyll *a*, which is located in LHC, and PSI/II[44,45]. The chlorophyll *a/b* ratio decreased in *bpg4-1* and *bpg4-2* compared with the WT plants

but increased in *BPG4-OX-2* and *BPG4-OX-10* (Fig. 2f). These results suggest that the development of LHC antennae size could be promoted in *BPG4* knockout plants and suppressed in *BPG4*-overexpressing plants. By electron microscopy observations of *BPG4* knockout and overexpression plants, an increase in the thylakoid membrane stacking number per grana in *bpg4-1* was observed (Fig. 2g, h). Fragmentation of thylakoid membranes of stroma lamellae and a decrease in stacking number were conversely observed in *BPG4-OX-2* (Fig. 2g, h). These results suggest that BPG4 plays important roles in regulating not only the endogenous chlorophyll contents but also chloroplast development, such as LHC size and grana structure.

*BPG4* has three homologous genes, *BGH1*, *BGH2*, and *BGH3*, in Arabidopsis, and high similarity in the region including the A_thal_3526 domain was detected between BPG4 and BGHs (Supplementary Fig. 4a), suggesting that BGHs may have redundant functions with BPG4 in the control of chloroplast development. First, to reveal whether *BPG4* and *BGHs* were expressed in the same tissues, tissue-specific expression of the *BPG4* family was analyzed by RT–PCR (Supplementary Fig. 4b). *BGH1* expression was detected only in siliques, *BGH2* was

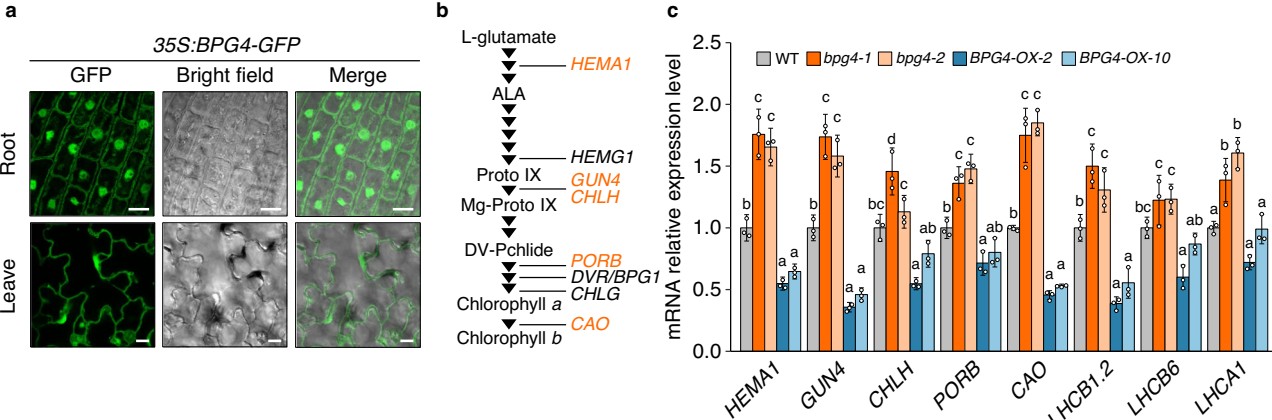

**Fig. 3 | BPG4 suppressed the expression of genes encoding chlorophyll biosynthesis-related enzymes and light-harvesting complex proteins in the nucleus. a** Confocal laser scanning microscopy images of root cells and epidermal cells of leaves of *35S:BPG4-GFP* transformants. Scale bars = 10 μm. The experiments were repeated at least three times independently and yielded similar results. **b** Chlorophyll biosynthesis pathway and chlorophyll biosynthesis-related genes in *Arabidopsis*. The arrows indicate each step. The genes include the following: glutamyl-tRNA reductase (*HEMA1*); protoporphyrinogen oxidase (*HEMG1*); magnesium chelatase cofactor (*GUN4*); magnesium chelatase H subunit (*CHLH*); protochlorophyllide oxidoreductase (*PORB*); 3,8-divinyl protochlorophyllide *a* 8-vinyl reductase (*DVR/BPG1*); chlorophyll synthase (*CHLG*); and chlorophyllide *a* oxygenase (*CAO*). The genes with orange are those whose expression was regulated by

BPG4. **c** Relative expression of genes encoding chlorophyll biosynthesis-related enzymes and light-harvesting complex proteins in WT (Col-0), *bpg4-1*, *bpg4-2*, *BPG4-OX-2*, and *BPG4-OX-10* grown on half-strength MS medium for 8 days. The genes include light-harvesting chlorophyll *a/b* binding protein 1.2 (*LHCB1.2*), light-harvesting complex photosystem II subunit 6 (*LHCB6*), and Photosystem I light-harvesting complex 1 (*LHCA1*). The relative expression levels were normalized to *GAPDH*. The means and SDs were obtained from three biological replicates. The different letters above the bars indicate statistically significant differences between the samples (one-way ANOVA: Tukey–Kramer's multiple comparisons test, $P < 0.05$), and groups with the same letters are not significantly different. Other gene expression levels are shown in Supplementary Fig. 7.

highly expressed in nongreen tissues such as roots and flower buds, and *BGH3* and *BPG4* were highly expressed in green tissues such as rosette leaves, which was almost consistent with genome-wide analysis published in Arabidopsis eFP Browser[46] (http://bar.utoronto.ca/efp//cgi-bin/efpWeb.cgi?dataSource=Klepikova_Atlas). Second, the expression of *BGH2* and *BGH3* in WT, *BPG4* knockout and overexpressing seedlings was analyzed, because the expression of homologous genes with redundant functions was often regulated in a feedback manner (Supplementary Fig. 4c). *BGH2* expression was not altered in *BPG4* knockout and overexpressing plants, while *BGH3* expression was increased in *bpg4-1* and *bpg4-2* compared with WT plants, and decreased in *BPG4-OX-2* and *BPG4-OX-10*. Next, *BGHs* overexpressing and knockout plants (*bgh1-1*, *bgh2-1*, *bgh2-2*, *bgh3-1*, and *bgh3-2*) were generated by transformation using the *35S* promoter and genome-editing based on CRISPR–Cas9 system, respectively (Supplementary Fig. 2b–e, Supplementary Fig. 5a–e). The cotyledons of the *BGH2-OX-2* and *BGH2-OX-11* seedlings were paler green than those of WT, similar to *BPG4-OX*, while *BGH3-OX-1* and *BGH3-OX-20* showed slightly paler green cotyledons than WT, which were weaker phenotypes than *BPG4-OX* (Supplementary Fig. 5b–e). The significant difference was not detected in WT, *bgh1-1*, *bgh2-1*, *bgh2-2*, *bgh3-1*, and *bgh3-2*, in contrast to *BPG4*-knockout plants (Supplementary Fig. 5a–e). Finally, *bpg4bgh2* and *bpg4bgh3* double mutants were generated by crossing *bpg4-1* and *bgh2-1* or *bgh3-1*, respectively (Supplementary Fig. 5f, g). The endogenous chlorophyll contents of *bpg4bgh2* and *bpg4bgh3* were not significantly different from that of *bpg4-1*. These results suggested that BGHs might have similar functions as BPG4 in the regulation of chloroplast development, but *BGH1* and *BGH2* might be expressed in different tissues from *BPG4*, and BGH3 function might be weaker than that of BPG4.

### BPG4 suppressed the expression of photosynthesis-associated nuclear genes in the nucleus

To elucidate how BPG4 suppresses chloroplast development and chlorophyll contents, the subcellular localization of the BPG4 protein was examined. Transgenic plants that expressed the BPG4 protein

fused to GREEN FLUORESCENT PROTEIN (GFP) driven by the *35S* promoter were generated, and the BPG4-GFP fluorescence signal was observed in the cytosol and nucleus (Fig. 3a). Moreover, we performed immunofluorescence staining using an anti-BPG4 antibody in *BPG4*-overexpressing plants (Supplementary Fig. 6). The fluorescent signal by an anti-BPG4 antibody was detected to overlap with the DAPI (4′,6-diamidino-2-phenylindole) signal. These results revealed that endogenous BPG4 protein was localized in the nucleus. Photosynthesis-associated proteins, such as chlorophyll biosynthesis-related enzymes, LHC subunits, components of PSI/II, and Rubisco, are localized in chloroplasts, while many important chloroplast proteins, such as chlorophyll biosynthesis-related enzymes and LHC, are encoded by the nuclear genome as photosynthesis-associated nuclear genes (*PhANGs*). Based on the observed localization of BPG4 in the nucleus, a possible function of BPG4 is thought to regulate the expression of *PhANGs* in the nucleus. A qRT–PCR analysis of 14 photosynthesis-associated genes in the WT, *BPG4* knockout plants, and *BPG4*-overexpressing plants was performed (Fig. 3b, c, Supplementary Fig. 7). The expression of 5 chlorophyll biosynthesis-related genes and 3 *LHC* subunit genes increased in *bpg4-1* and *bpg4-2* compared with the WT plants but decreased in *BPG4-OX-2* and *BPG4-OX-10* (Fig. 3b, c). These results suggest that BPG4 comprehensively suppresses the expression of *PhANGs* in the nucleus.

### BPG4 interacted with GLK transcription factors in the nucleus

To elucidate the detailed molecular mechanism through which BPG4 suppresses *PhANG* expression, the interactions of BPG4 with three major photosynthesis-associated factor families that function in the nucleus, HYPOCOTYL LONG 5 (HY5), CONSTITUTIVE PHOTO-MORPHOGENIC 1 (COP1), and GLKs, were analyzed by a yeast two-hybrid (Y2H) assay one by one (Fig. 4a, Supplementary Fig. 8). Interactions of BPG4 with GLK1 and GLK2 were detected (Fig. 4a), but interactions with HY5 and COP1 were not (Supplementary Fig. 8). GLK transcription factors are known to play major roles in the formation of the photosynthesis apparatus through their ability to promote the expression of *PhANGs* such as chlorophyll biosynthesis-related

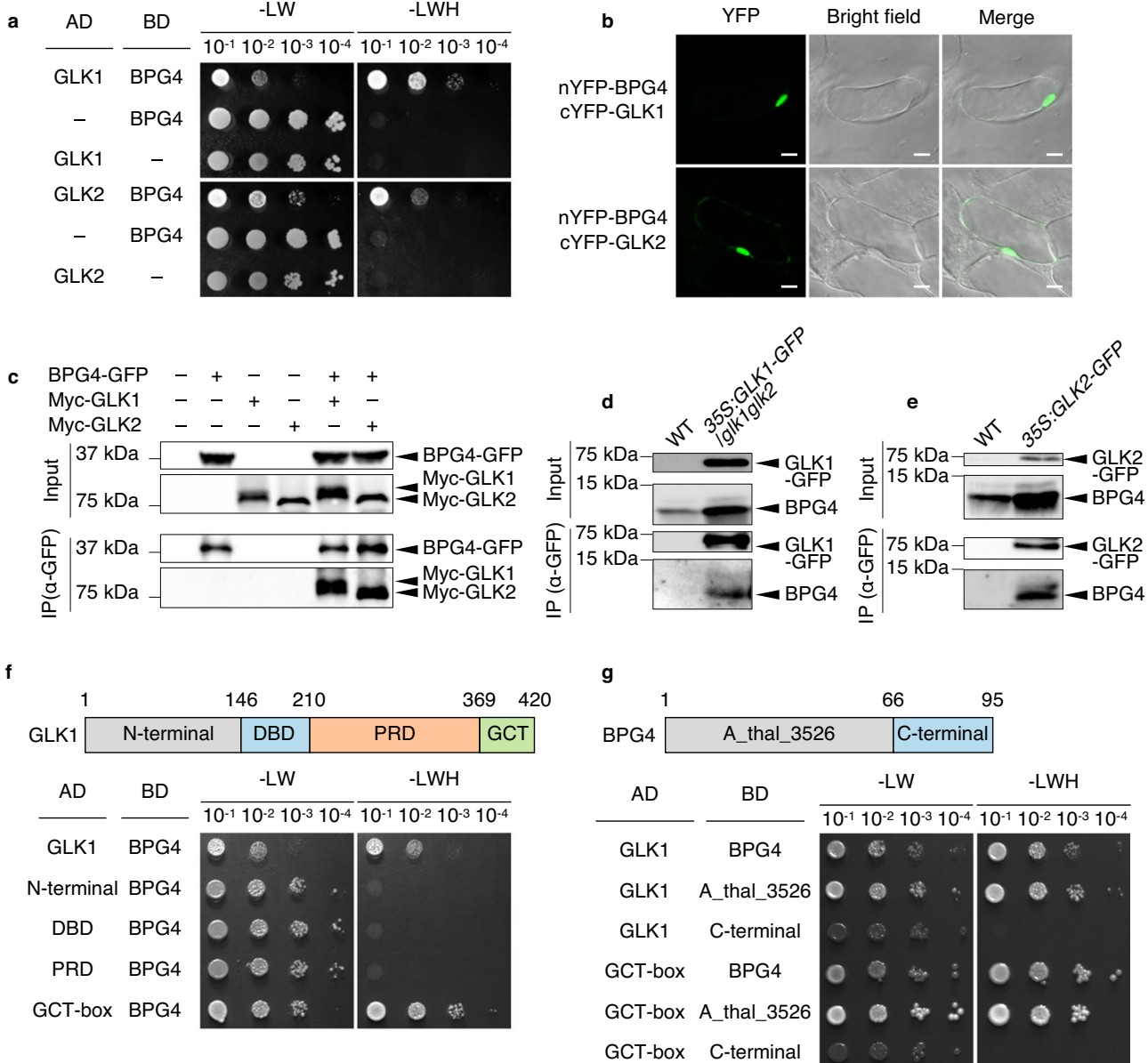

**Fig. 4 | BPG4 physically interacted with GLK transcription factors in the nucleus. a** Y2H assays to test the interactions of BPG4 with GLK1 and GLK2. The results of an analysis to test the interactions of BPG4 with other factors are shown in Supplementary Fig. 8. **b** BiFC assay results of the interactions of BPG4 (fused to the N-terminal fragment of YFP) with GLK1 and GLK2 (fused to the C-terminal fragment of YFP) in suspension-cultured Arabidopsis cells. Negative controls and confirmation of *nYFP* and *cYFP* expression in suspension-cultured cells are shown in Supplementary Fig. 9. Scale bar = 10 μm. The experiments were repeated at least three times independently and yielded similar results. **c** CoIP results of the interactions of BPG4 with GLK1 and GLK2. BPG4-GFP and 10x Myc-GLK1 or 10x Myc-GLK2 were transiently expressed in *Nicotiana benthamiana*. BPG4-GFP was immunoprecipitated using GFP-Trap A and immunoblotted using anti-GFP and anti-Myc antibodies. CoIP results of the interactions of endogenous BPG4 with GLK1 (**d**) and GLK2 (**e**) in Arabidopsis. WT (Col-0) and *35S:GLK1/glk1glk2* (**d**) or *35S:GLK2* (**e**) seedlings were grown on half-strength MS medium for 14 days. GLK1/2-GFP were immunoprecipitated using GFP-Trap A and immunoblotted using anti-GFP and anti-BPG4 antibodies. Y2H assays to identify the domain in GLK1 interacting with BPG4 (**f**) and the domain in BPG4 interacting with GLK1 (**g**). GLK1, full-length GLK1; N-terminal, aa 1–146; DNA-binding domain (DBD), aa 147–210; proline-rich domain (PRD), aa 210–369; GLK C-terminal domain (GCT-box), aa 369–420, BPG4, full-length BPG4; A_thal_3526, aa 1–66; C-terminal, aa 67–95. Negative controls are shown in Supplementary Fig. 9.

enzyme-encoding genes and LHC genes[6,9]. To analyze the interaction between BPG4 and GLKs in vivo, bimolecular fluorescence complementation (BiFC) assays in Arabidopsis suspension-cultured cells were performed in which BPG4 fused to the N-terminus of YELLOW FLUORESCENT PROTEIN (nYFP-BPG4) and GLK1/2 fused to the C-terminus of YFP (cYFP-GLK1/2) were used (Fig. 4b, Supplementary Fig. 9a, b). After coexpression of nYFP-BPG4 and cYFP-GLK1/2, the expression levels of *nYFP* and *cYFP* were confirmed via qRT–PCR (Supplementary Fig. 9a). YFP fluorescence signals in the nucleus were observed in cells coexpressing both nYFP-BPG4 and cYFP-GLK1/2 (Fig. 5b), while no signals were observed in cells expressing BPG4 or GLK1/2 alone (Supplementary Fig. 9b). Furthermore, coimmunoprecipitation (CoIP) assays were performed by transiently expressing BPG4 fused to GFP (BPG4-GFP) and GLK fused to Myc (Myc-GLK) in *Nicotiana benthamiana*, and interactions between BPG4 and GLK1/2 were confirmed (Fig. 4c). Finally, CoIP assays using an anti-BPG4 antibody also suggested that endogenous BPG4 protein interacted with GLK1/2-GFP in Arabidopsis (Fig. 4d, e).

GLKs contain four characteristic domains: an N-terminal domain consisting of an unstructured domain and a nuclear location signal

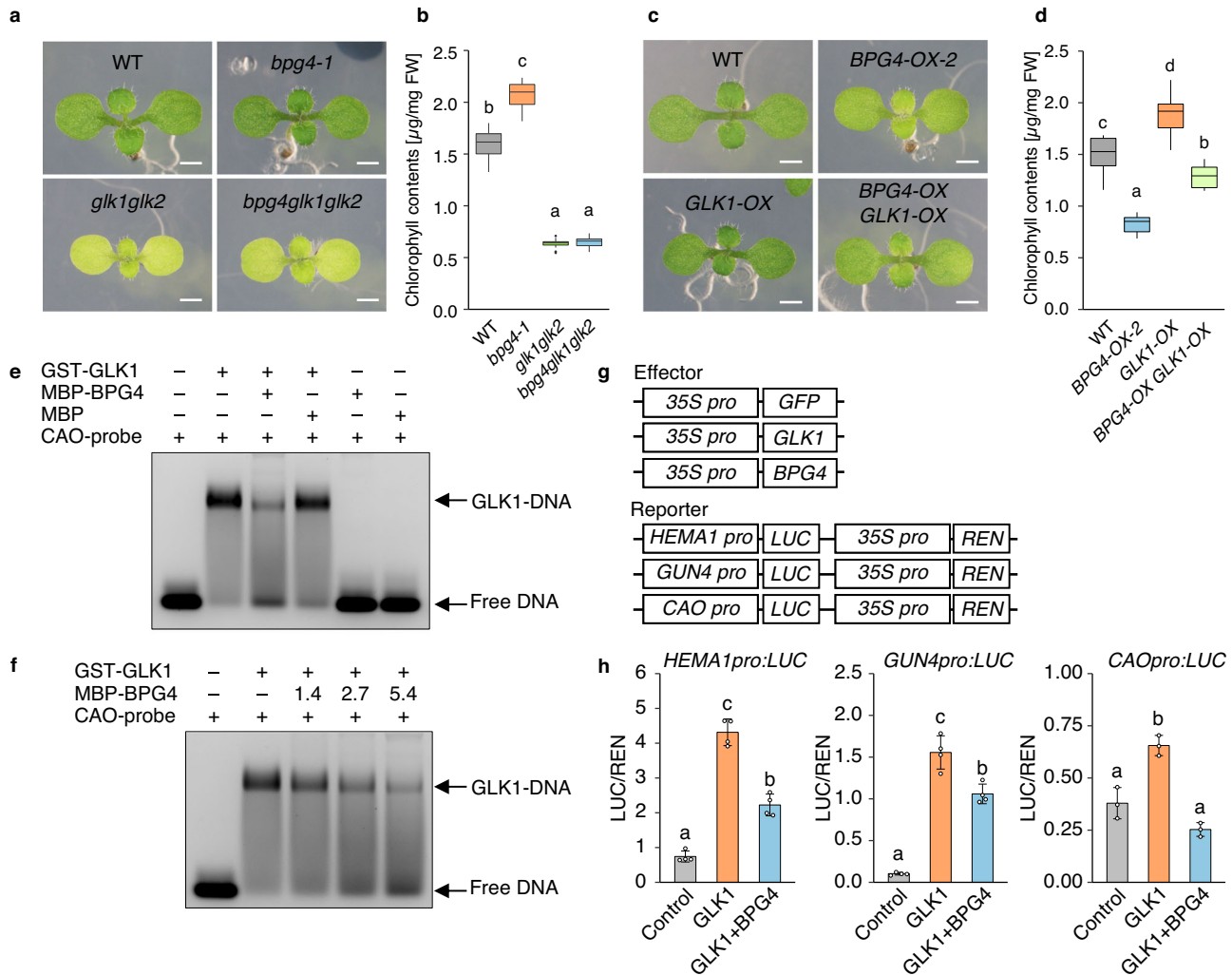

**Fig. 5 | BPG4 suppressed GLK1 transcriptional activity via inhibition of its binding to DNA.** Phenotypes (**a**, **c**) and endogenous contents of chlorophyll (**b**, **d**) in WT (Col-0), *bpg4-1*, *glk1glk2*, and *bpg4glk1glk2* seedlings (**a**, **b**) and WT, *BPG4-OX-2*, *GLK1-OX*, and *BPG4-OX GLK1-OX* seedlings (**c**, **d**) grown on half-strength MS medium for 8 days. Scale bars = 1 mm; *n* = 18 biologically independent samples. The center of the boxplot is denoted by the median, a horizontal line dividing the box into two equal halves. The bounds of the box are defined by the lower quartile (25th percentile) and the upper quartile (75th percentile). The whiskers extend from the box and represent the data points that fall within 1.5 times the IQR from the lower and upper quartiles. Any data point outside this range is considered an outlier and plotted individually. The different letters above the bars indicate statistically significant differences between the samples (one-way ANOVA: Tukey–Kramer's multiple comparisons test, *P* < 0.0001), and groups with the same letters are not significantly different. **e**, **f** EMSA experiments of GST-GLK1 or that cotreated with MBP or MBP-BPG4 with a FAM-labeled ds DNA containing the GLK1-binding site in the *CAO* promoter region. The numbers 1.4, 2.7, and 5.4 in (**f**) indicate the molar ratios of MBP-BPG4 versus GST-GLK1. The probe sequences used are shown in Supplementary Fig. 10. The experiments were repeated at least three times independently and yielded similar results. **g**, **h** Results of a transient assay to analyze the effects of BPG4 on GLK1 transcriptional activity on the *HEMA1*, *GUN4*, and *CAO* promoters in WT protoplasts. The control (GFP), GLK1 (GLK1 + GFP), or GLK1 + BPG4 effectors were coexpressed with each reporter. Relative LUC activity (LUC/REN) indicates the level of *HEMA1*, *GUN4*, and *CAO* promoter activity. Data are presented as the means ± SDs. *n* = 4, 4, and three biologically independent samples for *HEMA1pro:LUC*, *GUN4pro:LUC*, and *CAOpro:LUC*, respectively. The different letters above the bars indicate statistically significant differences between the samples (one-way ANOVA: Tukey–Kramer's multiple comparisons test, *P* < 0.01), and groups with the same letters are not significantly different.

(N-terminal), a Myb-DNA-binding domain (DBD), a proline-rich domain (PRD), and a C-terminal domain referred to as the GCT-box[9,20,47,48]. BPG4 contains an evolutionarily conserved domain, A_thal_3526. To map the subdomains of GLK1 and BPG4 required for their interaction, we carried out Y2H assays using fragments separated into each subdomain and revealed that the A_thal_3526 domain of BPG4 interacted with the GCT-box of GLK1 (Fig. 4f, g, Supplementary Fig. 9c, d).

### BPG4 suppressed GLK1 transcriptional activity via inhibition of DNA-binding ability

To investigate the genetic interaction of BPG4 with GLKs, *bpg4glk1glk2* triple mutants were generated by crossing *bpg4-1* and *glk1glk2*.

Compared with those of the WT, the cotyledons of *bpg4-1* were darker green, while those of *glk1glk2* and *bpg4glk1glk2* were paler green, and the endogenous content of chlorophylls in the *bpg4glk1glk2* triple mutants was the same as that in *glk1glk2* (Fig. 5a, b). Furthermore, double transformants overexpressing *BPG4* and *GLK1* (*BPG4-OX GLK1-OX*) were generated by crossing *BPG4-OX-2* and *GLK1-OX*. With respect to the single transformants, compared with those of the WT plants, the cotyledons of *BPG4-OX-2* were paler green, and those of *GLK1-OX* were darker green. With respect to *BPG4-OX GLK1-OX*, the pale green cotyledons of *BPG4-OX-2* were partly rescued by overexpressing *GLK1* (Fig. 5c, d). These results suggest that *glk1glk2* and *GLK1-OX* are epistatic to *bpg4-1* and *BPG4-OX-2* and that GLKs act downstream of BPG4 in the control of chlorophyll biosynthesis.

To elucidate the molecular effect of BPG4 on GLKs in vitro, we performed electrophoretic mobility shift assays (EMSAs) in which the promoter sequence of the GLK1 target gene *CAO* and recombinant GLK1 protein with or without BPG4 were used (Fig. 5e, f, Supplementary Fig. 10). The signal intensity of the GLK1-DNA complex decreased in the presence of BPG4 protein compared with that in the absence of BPG4 (Fig. 5e), and the degree of signal reduction increased with the amount of BPG4 added (Fig. 5f). These results suggest that the DNA-binding ability of GLK1 is suppressed by the interaction with BPG4 in a dose-dependent manner. Finally, to analyze the effect of BPG4 binding on GLK transcriptional activity in vivo, we generated a luciferase (LUC) reporter fused to the promoter region of the GLK target genes *HEMA*, *GUN4*, and *CAO* and the effectors *35S:BPG4* and *35S:GLK1* and performed LUC-based transient transactivation assays (Fig. 5g, h). The relative signal intensity of firefly LUC to Renilla luciferase (REN) of each reporter increased when the *35S:GLK1* effector was transfected without the *35S:BPG4* compared with the control. The intensity decreased when the *35S:GLK1* effector was cotransfected with *35S:BPG4* effector in comparison to only *35S:GLK1* transfection (Fig. 5h).

To investigate the effects of *GLK1* overexpression in the absence of its inhibitor, BPG4, on *PhANG* expression and chloroplast development, *bpg4-1 GLK1-OX* was generated by crossing *bpg4-1* and *GLK1-OX*. *bpg4-1 GLK1-OX* showed similar levels of chlorophyll content and *PhANG* expression to *bpg4-1* (Supplementary Fig. 11a–c). *GLK1* expression in *GLK1-OX* was 21.9-fold higher than that in WT, but only 2.4-fold higher in *bpg4-1 GLK1-OX* (Supplementary Fig. 11d), which would result in similar chlorophyll levels in *bpg4-1* and *bpg4-1 GLK1-OX*. The results imply that overexpressed *GLK1* without *BPG4* might be regulated by some unknown mechanisms. Taken together, these results suggest that BPG4 interacts with GLK transcription factors to suppress their transcriptional activities via inhibition of their DNA-binding ability.

## BPG4 expression was suppressed by BR signaling via the BR signaling master transcription factor BES1

As the *bpg4-1D* mutants were isolated by their pale green cotyledon phenotypes even in the presence of Brz, which causes a BR deficiency, BPG4 is thought to play an important role in BR signaling. To reveal the relationship between BPG4 and BR signaling, the effect of Brz treatment on *BPG4* expression and BPG4 protein accumulation in the light was analyzed. *BPG4* expression was increased by Brz treatment (Fig. 6a), and BPG4 protein was also accumulated (Fig. 6b). Conversely, 3 h of brassinolide (BL) treatment decreased *BPG4* expression compared with mock treatment (Fig. 6c). These results suggest that BR signaling suppresses *BPG4* expression.

Next, to clarify the mechanism of how *BPG4* expression is regulated in BR signaling, *BPG4* expression in several BR biosynthesis and signaling mutants was analyzed. *BPG4* expression in the BR signaling master transcription factor BRI1-EMS-SUPPRESSOR 1 (BES1) gain-of-function mutant *bes1-D* decreased in comparison with that in WT plants (Fig. 6d). In contrast, *BPG4* expression in the BR biosynthesis-deficient mutant *det2*, BR receptor-deficient mutant *bri1-5*, and BR signaling kinase BIN2 gain-of-function mutant *bin2-1* increased compared with that in the WT plants (Supplementary Fig. 12). Given that BES1 functions most downstream of BR signaling in the four BR mutants analyzed, the suppression of *BPG4* expression by BR signaling could be related to BES1.

To verify the possibility of direct regulation to *BPG4* expression by BES1, the *BPG4* promoter region was analyzed, and two G-box (CACGTG), the sequences of which are typical BES1-binding motifs[19], were detected 288 bp (G-box 1) and 180 bp (G-box 2) upstream of the start codon (ATG) (Fig. 6e). Then, we performed a chromatin immunoprecipitation (ChIP) assay for *BES1pro:BES1-GFP* transformants with an anti-GFP antibody and detected the binding of BES1-GFP onto the *BPG4* promoter region containing two G-box sites (Fig. 6f). Next, we

generated a dual-LUC reporter fused to the promoter region of *BPG4* and the effectors *35S:BES1* and *35S:BES1-VP64* (BES1 fused with the tetrameric repeat of herpes simplex VP16's minimal activation domain (VP64)[49], a strong transcriptional activation domain in plant cells[50,51]) and performed LUC-based transient transactivation assays (Fig. 6g, h). The relative signal intensity of LUC to REN decreased when the *35S:BES1* effector was transfected compared with the control, but increased when *35S:BES1-VP64* was transfected (Fig. 6h). Furthermore, we carried out EMSAs using a recombinant protein of the BES1 DNA-binding domain (DBD) and a DNA probe of either G-box 1 or G-box 2, including the 24 bp sequences around G-box (Fig. 6e, i, Supplementary Fig. 13a, b). The BES1 DBD-DNA complex was observed with probes for both G-box 1 and G-box 2, but the intensity of the probe for G-box 2 was much stronger than that for G-box 1 (Fig. 6i, Supplementary Fig. 13b). After we mutated the G-box sequence (CACGTG) to AAAAAA, BES1 DBD-DNA complexes were no longer detected (Fig. 6i). Finally, we verified the effect of BES1 binding specificity to the G-box on regulating *BPG4* expression, using LUC-based transient assays with mutated G-box sites in the *BPG4* promoter region (Fig. 6j, k). BES1-VP64 elevated the LUC/REN intensities of the *BPG4* reporter without mutations (WT) and with only G-box 1 mutation (Mu-1) compared to the control but could not increase those with G-box 2 mutation (Mu-2) and double mutation (Mu1&2). These results suggest that BES1 directly binds to only G-box 2 in the *BPG4* promoter region and suppresses *BPG4* expression as an event involved in BR signaling.

In immunoblot analysis using an anti-BPG4 antibody, endogenous BPG4 protein was detected in two forms, one major band and another upper shifted band (Figs. 2c and 6b). Since splicing variants of BPG4 have not been recognized according to the latest database (Araport11), it was anticipated that these two forms might consist of post-translationally modified BPG4. Phosphorylation is widely recognized for its ability to impart a negative charge to proteins, thereby shifting their mobility during electrophoresis. First, to determine the possibility for phosphorylation of endogenous BPG4 protein, we treated lambda protein phosphatase (λPP) to the extracts from WT and *BPG4-OX-2*, and performed immunoblot analysis using an anti-BPG4 antibody (Supplementary Fig. 14a). Both band signals of BPG4 were not altered by λPP treatment. Additionally, the interaction of BPG4 with BIN2, a major cytoplasmic-type kinase in BR signaling, was not detected in the Y2H assay (Supplementary Fig. 14b). Bikinin, a BIN2 inhibitor, did not have obvious effects on the status of the BPG4 protein (Supplementary Fig. 14c). These results suggest that the two-band signal of the endogenous BPG4 protein were not regulated by phosphorylation. Then, to investigate the status and stability of the BPG4 protein under BR signaling, immunoblot analysis using an anti-BPG4 antibody was performed. BL and Brz treatment did not influence the status and accumulation of BPG4 protein (Supplementary Fig. 15a–d). Endogenous BPG4 protein levels were altered in *bin2-1*, *BRI1-OX*, *gsk3 quadruple* mutants (*bin2 bil1 bil2 ATSK13RNAi*), and *bri1-5*, but these results would depend on the regulation of *BPG4* mRNA expression by BES1 (Supplementary Fig. 15e, f). In summary, it appears that BPG4 is regulated at the mRNA level by BR signaling, rather than at the protein level.

## BPG4 was expressed specifically in photosynthetic tissues in the light and was regulated by the circadian rhythm

Compared with WT plants, the BR-deficient mutant *det2*, and the BR signaling mutants *bri1-5* and *bin2-1* grown in the light increased endogenous chlorophyll contents and promoted development of thylakoid grana structure in chloroplasts[20]. The phenotype of Brz-treated WT plants is also the same as that of these mutants[34]. In contrast, brassinolide (BL) treatment decreases the chlorophyll content in a dose-dependent manner[20], and the chlorophyll content was shown to decrease in the BR signal-accelerating mutant *bes1-D*[19]. Based on the consideration from these knowledge, it is thought that BR signaling generally suppresses chlorophyll biosynthesis and chloroplast

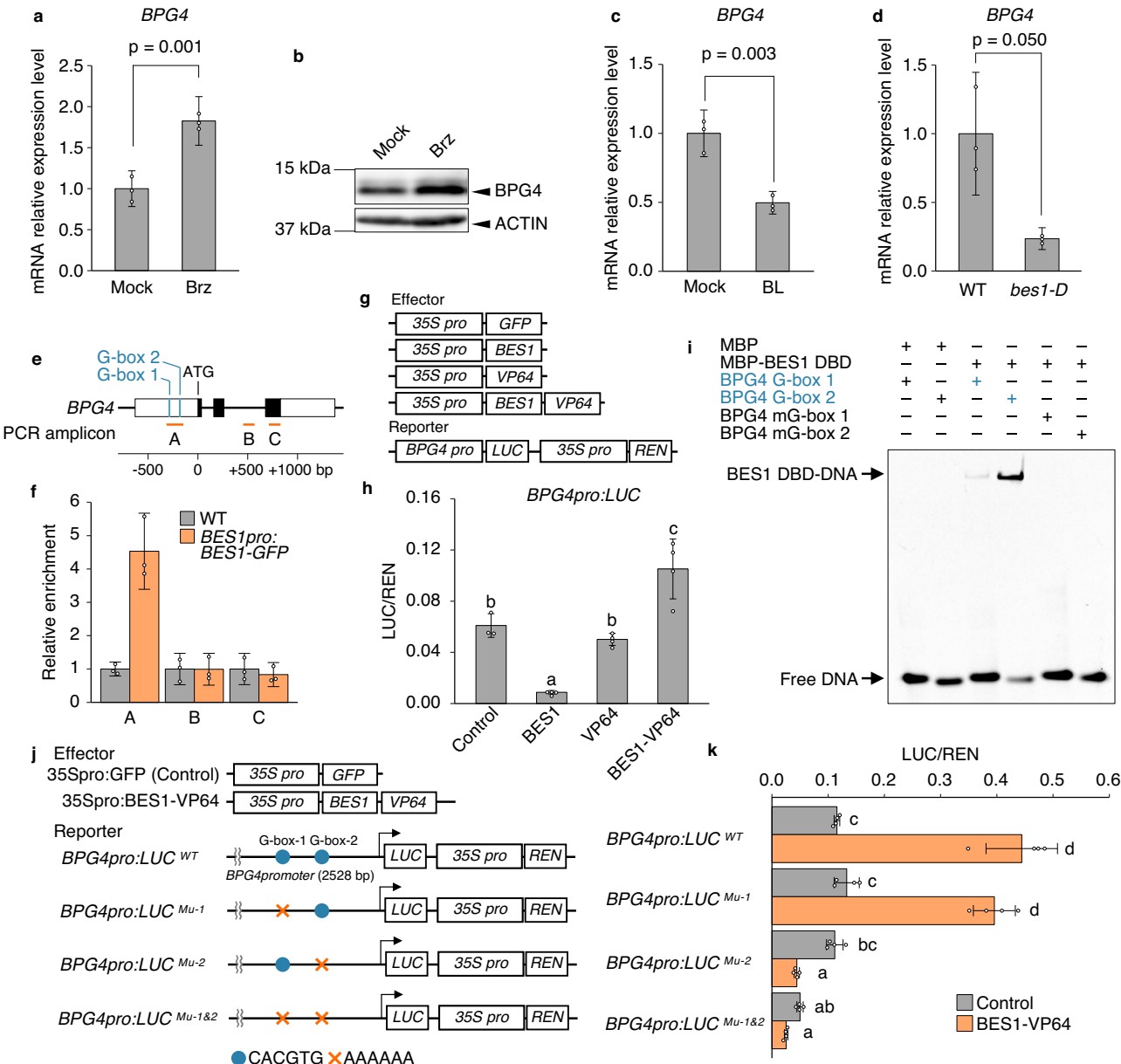

**Fig. 6 | *BPG4* expression was induced by BR deficiency and suppressed by BES1 bound to the *BPG4* promoter region.** Analysis for relative expression of *BPG4* (**a**, **c**, **d**), and detection of BPG4 proteins using an anti-BPG4 antibody (**b**). WT seedlings were grown on half-strength MS medium with 3 μM Brz or the same volume of DMSO solvent (mock) for 7 (**a**) or 9 (**b**) days. WT seedlings were grown for 12 days, and treated with 500 nM BL, or mock for 3 h (**c**). WT and *bes1-D* were grown on half-strength MS medium for 8 days (**d**). The relative expression levels were normalized to *GAPDH*. The means and SDs were obtained from three biological replicates. Statistical analysis was performed via two-sided Student's *t* test (**a**, **c**) or two-sided Welch's *t* test (**d**). **e** A diagram showing the structure of the *BPG4* gene. The black and white boxes indicate exon regions, and UTRs, respectively. G-box in the *BPG4* promoter and the DNA fragments amplified in (**f**) are marked with blue or orange lines, respectively. ATG denotes the start codon. **f** ChIP analysis of the binding of BES1 to the *BPG4* promoter. Data are presented as the means ± SDs of three technical replicates. Schematic drawing of various constructs used in the transient assay (**g**), and relative LUC activity (LUC/REN) driven by *BPG4* promoter in WT protoplasts (**h**). GFP is used as a control. Data are presented as the means ± SDs of biologically independent samples. *n* = 4 except for Control (*n* = 3). The different letters above the bars indicate statistically significant differences between the samples (one-way ANOVA: Tukey–Kramer's multiple comparisons test, *P* < 0.01). **i**, EMSA experiment results of MBP-fused BES1 DBD with a DNA probe containing a G-box or mG-box (mutated G-box) in the *BPG4* promoter region. Schematic drawing of various constructs used in the transient assay (**j**), and relative LUC activity driven by *BPG4* promoter with or without G-box mutations in WT protoplasts (**k**). The means and SDs were obtained from four biological replicates. The different letters above the bars indicate statistically significant differences between the samples (one-way ANOVA: Tukey–Kramer's multiple comparisons test, *P* < 0.05).

development in the light. *BPG4* expression was suppressed by BES1 and BL treatment (Fig. 6c, d), although BPG4 suppressed chlorophyll biosynthesis and chloroplast development (Fig. 2a–h). Therefore, in terms of BR signaling-mediated regulation of chloroplast development, the expression of *BPG4* appeared to be inconsistent with the protein function of BPG4. To resolve this contradiction and clarify the actual

role of BPG4 in regulating chloroplast development through BR signaling and the physiological significance of BPG4 for plant growth, we analyzed tissue-specific *BPG4* expression and the effect of light, another chloroplast regulatory signal, on *BPG4* expression.

First, to analyze the tissue-specific expression of *BPG4*, transgenic plants that expressed β-glucuronidase (GUS) fused to the *BPG4*

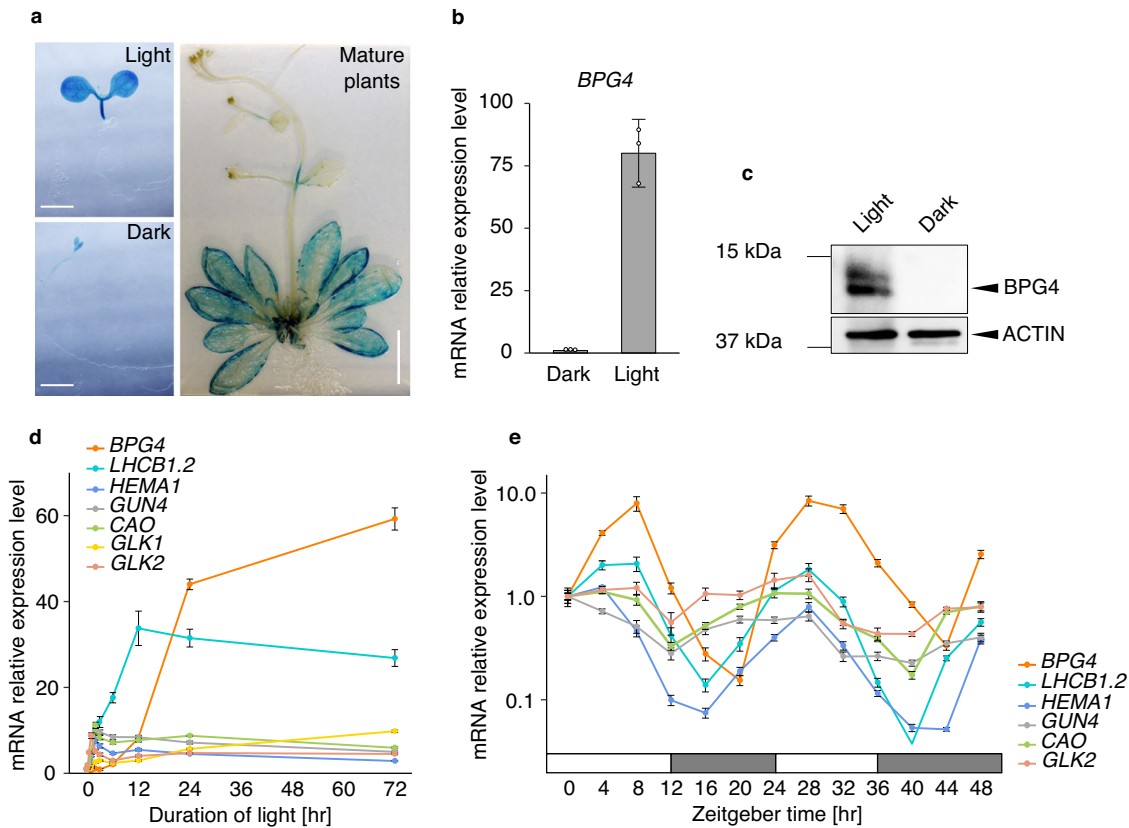

**Fig. 7 | BPG4 expression was induced by light and was regulated by the circadian rhythm. a** GUS expression pattern of *BPG4promoter:GUS* in seedlings grown on half-strength MS in the light or dark for 7 days (left) or plants grown on soil for 30 days (right). Scale bar = 2 mm (left) or 2 cm (right). The experiments were repeated at least three times independently and yielded similar results. Relative expression of *BPG4* (**b**) and detection of BPG4 protein via anti-BPG4 antibody (**c**) in WT (Col-0) seedlings grown on half-strength MS medium under the indicated conditions. The plants were grown under continuous light conditions or dark conditions for 7 days. The relative expression levels were normalized to *EUKARYOTIC TRANSLATION INITIATION FACTOR 4a1* (*eIF4a*). Data are presented as the means ± SDs of three technical replicates. ACTIN was used as a loading control. **d**, **e** Relative expression of *BPG4*, *GLKs*, and *PhANGs* in WT seedlings grown on half-strength MS medium under the indicated conditions. In (**d**), the plants were grown under continuous light for 0.25, 0.5, 1, 2, 3, 6, 12, 24, or 72 h after growing in the dark for 4 days. An enlarged view of the graph from 0 to 12 h is shown in Supplementary Fig. 16. In (**e**), WT plants grown on soil under LD conditions for 24 days were transferred to continuous light (LL) conditions for 48 h, and subsequently collected every 4 h in 48 h. The white boxes indicate the day time, and the gray boxes indicate the night time of the LD cycle before the LL conditions were imposed. The relative expression levels were normalized to *eIF4a* and represented as *n*-fold changes relative to the value of the sample at 0 h (**d**) or ZT0 (**e**). The *y*-axis in (**e**) is on a logarithmic scale. Data are presented as the means ± SDs of at least three technical replicates. Precise values of replicates (*n*) for each data are provided in Source data file. Each qRT–PCR was performed at least three times independently and yielded similar results.

promoter region were generated (Fig. 7a). As shown in Fig. 7a, GUS staining was clearly observed in the cotyledons of the light-grown seedlings but was weak in those of the dark-grown seedlings, and GUS staining was prevalent in the rosette leaves of mature plants. The differences in *BPG4* expression and BPG4 protein accumulation in the light and dark were confirmed by qRT–PCR and immunoblot analysis (Fig. 7b, c). Second, for a detailed analysis of the *BPG4* expression pattern in response to light exposure, we performed qRT–PCR on WT plants exposed to light after being grown in the dark for 4 days (Fig. 7d, Supplementary Fig. 16). The expression of *GLKs* and *PhANGs*, such as *LHC* genes that were directly regulated by GLKs, was quickly induced in response to light within 2 h and peaked within 12 h, while *BPG4* expression was induced slowly and was more delayed in response to light (induction began at 6 h) but continued to increase until 72 h after light exposure (Fig. 7d, Supplementary Fig. 16). Finally, to clarify that the expression of *BPG4* coincides with the circadian rhythm, qRT–PCR was carried out on WT plants exposed to continuous light for 48 h after having been grown under light/dark (LD) cycles (light for 12 h/darkness for 12 h) for 3 weeks (Fig. 7e). *BPG4* expression was induced during the day and was the highest at Zeitgeber time (ZT) 4 to ZT 8 and from ZT 28 to ZT 32. Conversely, *BPG4* expression was suppressed at night and

was lowest at ZT 20 and ZT 44. The expression of *GLK2* and *PhANGs* was highest at ZT 0 to ZT 4 and from ZT 24 to ZT 28, and was lowest at ZT 12 to ZT 16 and from ZT 36 to ZT 40. Thus, *BPG4* expression was strictly regulated by the circadian rhythm, and was slightly delayed to the expression of *GLK2* and *PhANGs* (Fig. 7e). The possible biological significance of the delay of *BPG4* expression in comparison with *PhANGs* expression is considered in the Discussion.

### BPG4 deficiency caused increased ROS generation and a reduction in photosynthetic activity under excessive high-light conditions

In Fig. 7, *BPG4* expression was promoted by not only BR-deficient conditions but also light. As both BR deficiency and light promote chlorophyll biosynthesis and chloroplast development, the dynamics of *BPG4* expression are expected to be opposite and inconsistent with those of BPG4 protein function to suppress chlorophyll biosynthesis and chloroplast development. Nevertheless, the viewpoint of maintaining homeostasis for chloroplasts would be able to resolve this contradiction and hypothesize the actual role and physiological significance of BPG4 in plant growth. Chlorophylls in PSI/II and LHC absorb light energy, become excited, and drive photosynthesis. Under

several environmental stress conditions such as high light and low temperature, absorption of excess light energy causes the generation of reactive oxygen species (ROS) and inhibits photosynthetic activity, a phenomenon called photoinhibition[52,53]. Uncontrolled and excessive accumulation of free chlorophyll and its precursors also leads to ROS generation[54,55]. The generated ROS can cause membrane oxidation, protein breakdown, and cell death. Therefore, tight and optimized regulation of chlorophyll biosynthesis and LHC size is necessary to avoid ROS generation and drive efficient photosynthesis. Recently, it has been reported that excessive accumulation of chlorophyll precursors and LHCP by overactivation of GLKs accelerates ROS generation, causes photoinhibition, and leads to cell death[48,56]. Thus, fine and careful regulation of GLK activity would be essential for suppressing ROS generation and sustaining healthy photosynthesis. Recent reports also suggested that the expression of *GLKs* and their activity are enhanced by both light and BR deficiency[19,20]. Therefore, we considered that *BPG4* mRNA expression was activated by light and BR deficiency, concurrent with the activation of GLKs by light and BR deficiency, and BPG4 plays an important role in suppressing the excessive activity of GLKs by involving light and BR signaling, maintaining suitable amounts of endogenous chlorophyll contents, optimizing chloroplast development, and avoiding ROS generation.

We considered that this hypothesis for the physiological significance of BPG4 could be verified by investigating whether BPG4 function contributes to the suppression of ROS generation induced by excessive light energy absorption. We first performed histochemical staining using 3,3-diaminobenzidine (DAB) and nitro blue tetrazolium (NBT) for the detection of ROS generation and measured the amount of hydrogen peroxide ($H_2O_2$) and superoxide ($O_2^{\cdot-}$), representative ROS in plants, in WT plants, *BPG4* knockout plants, and *BPG4*-overexpressing plants after induction of ROS generation by exposure to excessive high-light (Fig. 8a, b). Induction of $H_2O_2$ and $O_2^{\cdot-}$ generation caused by high-light exposure was observed in each plant, and the amounts of $H_2O_2$ and $O_2^{\cdot-}$ were observed to be more highly increased in *bpg4-1* and *bpg4-2* than in the WT plants, while they were less increased in *BPG4-OX-2* and *BPG4-OX-10* (Fig. 8a, b). By quantifying the $O_2^{-}$ amount, the amount of $O_2^{-}$ under high-light conditions was lower in *BPG4-OX-2* than in the WT plants, and was higher in *bpg4-1* (Fig. 8c, d). These results suggest that BPG4 might suppress chloroplast development via GLKs, thereby helping reduce ROS generation induced by the absorption of excessive light energy under high-light conditions.

Finally, to investigate the function of BPG4 in photosynthesis activity, we analyzed the maximum quantum yield of PSII (Fv/Fm), a sensitive indicator of plant photosynthetic performance, of the WT plants, *BPG4* knockout plants, and *BPG4*-overexpressing plants after plants were transferred to high-light conditions (900 μmol photons $m^{-2} s^{-1}$) from normal light conditions (90 μmol photons $m^{-2} s^{-1}$) (Fig. 8e). The Fv/Fm values of each plant decreased with increasing high-light exposure duration, suggesting that photoinhibition occurred in each plant. In detail, the Fv/Fm value of *bpg4-1* was more strongly decreased by high-light exposure than that of the WT plants, while that of *BPG4-OX-2* remained at a level higher than that of the WT plants (Fig. 8e). Taken together, these results suggest that BPG4 helps maintain the photosynthetic activity that is suppressed by excessive light energy absorption. Therefore, BPG4 might be an important homeostasis factor to maintain effective and healthy photosynthesis in plants.

## Discussion

In this report, we isolated a gain-of-function mutant, *bpg4-1D*, as pale green−leaf phenotype that was insensitive to Brz, which induced greening, and identified *BPG4* (Fig. 1a–c, Supplementary Fig. 1). Currently, we are considering that BPG4 acts as a regulator of chloroplast development and homeostasis by suppressing GLK activity and

involving light and BR signaling. *BPG4* is widely conserved in land plants, and BPG4 had an evolutionarily conserved domain, A_thal_3526, which was conserved in not only land plants but also green algae, stramenopiles, discoba, and amoebozoa (https://www.ebi.ac.uk/interpro/entry/pfam/PF09713/) (Fig. 1d, e). Since the A_thal_3526 domain was required for the interaction with GLK1 (Fig. 4g), the domain could play a critical role in BPG4 function. Recently, a *BPG4* homologous gene in rice was identified as DEEP GREEN PANICLE 1, which reduced chlorophyll biosynthesis in panicles by repressing OsGLK1 activity[57], suggesting that the function of BPG4 is conserved in land plants.

In Arabidopsis, there are three *BPG4* homologs, namely *BPG4 Homologous genes* (*BGHs*), (Supplementary Figs. 2, 4, and 5). Our findings demonstrate that *BGH2-OX* and *BGH3-OX* showed a pale green phenotype, resembling that caused by *BPG4-OX* (Supplementary Fig. 5), which is consistent with a previous report by ref. 40. *BGH3* expression was predominantly observed in the same tissues as *BPG4* while *BGH1* and *BGH2* were not. Only *BGH3* expression was regulated in a feedback manner with *BPG4* expression (Supplementary Fig. 4). Previous research has reported that *BGH2/RPGE1* represses the expression of photosynthetic genes, promotes amyloplast development in the endodermis, and influences hypocotyl negative gravitropism downstream of PIFs[40]. According to the gene coexpression database[58] (ATTED-II; https://atted.jp/), *BPG4* and *BGH3* exhibited coexpression with *PhANGs*, but *BGH1* and *BGH2* did not. Based on these results, we considered that BGHs would largely share a similar function to BPG4 in the suppression of GLKs, but *BGH1* and *BGH2* were expressed in different tissues from *BPG4*, regulating plastid development in a different role from BPG4. Although BGH3 appeared to function in the same tissues as BPG4, *bgh3-1* and *bpg4bgh3* did not display markedly distinct phenotypes from the wild type (WT) and *bpg4-1*, respectively. This result suggests that functionality of BGH3 might be weaker than that of BPG4. Further detailed analysis for similarity and independency within the BPG4 family will be conducted in future investigations.

BPG4 was considered to be a negative regulator of chlorophyll biosynthesis and chloroplast development (Fig. 2a, d–h). The molecular function of BPG4 starts by the interactions with GLK transcription factors that play critical roles in chloroplast development (Fig. 4a–g). The interaction of BPG4 with GLKs inhibited the DNA-binding activity of GLKs (Fig. 5e–h), reduced the expression of *PhANGs* such as chlorophyll biosynthesis-related genes and *LHC* genes in the nucleus (Fig. 3a–c), and led to a decrease in chlorophyll biosynthesis and LHC sizes in chloroplasts (Fig. 2a, d–h). Generally, PSII and LHCII are thought to be localized on grana, which are multilayered structures in chloroplasts. LHCII accounts for approximately one-third of the total protein in plant thylakoids, binds to both chlorophyll *a* and *b*, and forms trimers, which play central roles in granum adhesion[59]. Thus, deficiency of *BPG4* in *bpg4-1* and *bpg4-2* caused the activation of GLKs, increasing chlorophyll *a* and *b* biosynthesis, increasing the accumulation of LHCII, and, finally, accelerating chloroplast development with more thylakoid stacks per grana.

BPG4 inhibited the DNA-binding and transcriptional activity of GLK1 via direct interaction with the GCT-box, not the DBD (Fig. 4f). The GCT-box was identified as a unique domain in the GLK family, and has been reported to be associated with homo- or heterodimerization in maize and rice[47]. Given that Turnip Yellow Mosaic Virus P69 interacted with the region containing GCT-box of GLK1[60] and thereby suppressed the transcriptional activity of GLK1, the GCT-box is considered to play critical roles in the DNA-binding and transcriptional activity of GLK1. To reveal how the GCT-box governs GLK transcriptional activity and how BPG4 suppresses the DNA-binding activity of GLK1 via interaction with the GCT-box, further research is needed.

In recent studies, some phytohormone signaling pathways, including BRs, have been reported to coordinate GLK transcriptional activity[20,48,56]. Downstream of salicylic acid (SA) signaling, SIGMA

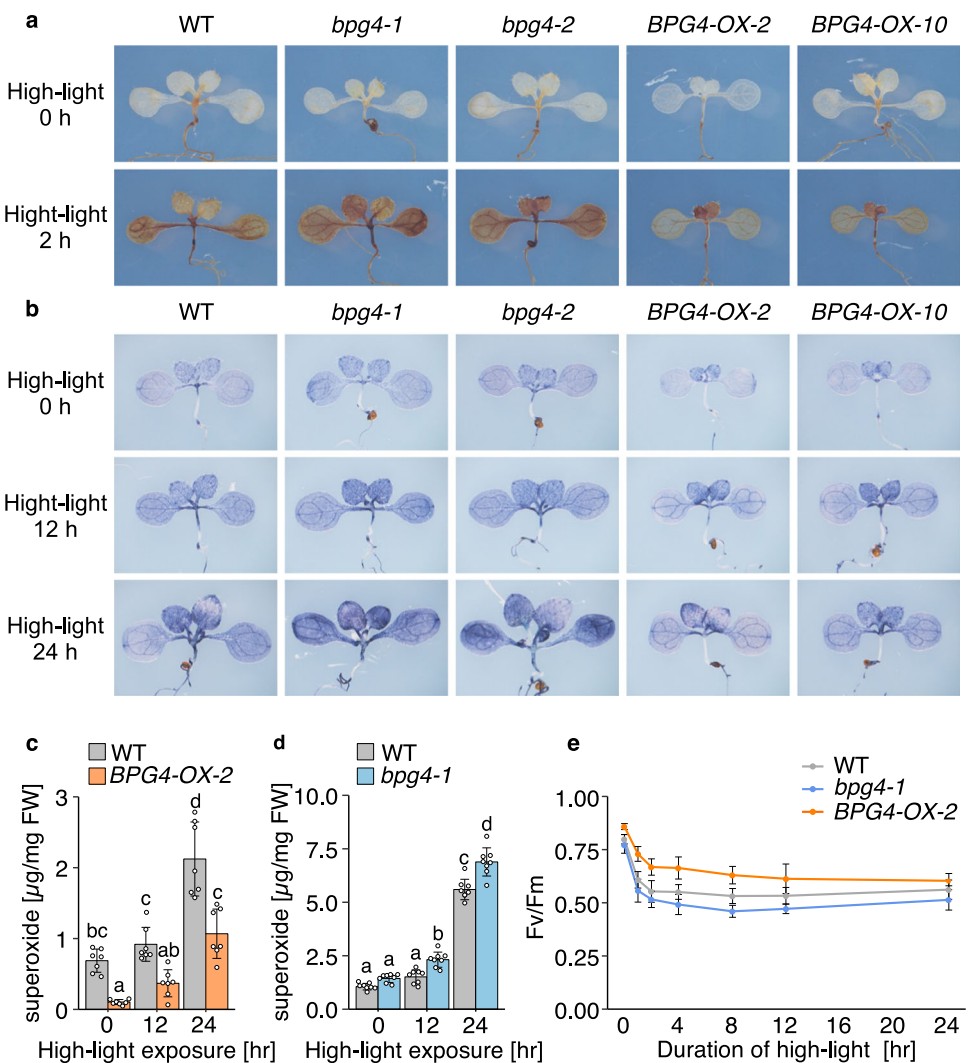

**Fig. 8 | *BPG4* deficiency caused increased ROS generation and reduced photosynthetic activity under excessive high-light conditions.** Visualization of $H_2O_2$, as detected by DAB staining (**a**) and $O_2^{·-}$ radicals, as detected by NBT staining (**b**). WT (Col-0), *bpg4-1*, *bpg4-2*, *BPG4-OX-2*, and *BPG4-OX-10* seedlings grown on half-strength MS medium under a growth light (90 μmol photons m$^{-2}$ s$^{-1}$, long-day conditions) for 7 days were transferred to high-light conditions (1200 μmol photons m$^{-2}$ s$^{-1}$) for 2 h (**a**). The leaves were infiltrated with DAB before exposure to high light. Seedlings grown on half-strength MS medium under a growth light (90 μmol photons m$^{-2}$ s$^{-1}$, long-day conditions) for 8 days were transferred to high-light conditions (800 μmol photons m$^{-2}$ s$^{-1}$, continuous light) for 12 or 24 h (**b**). Quantification of endogenous contents of $O_2^{·-}$ in WT and *BPG4-OX-2* (**c**) and in WT and *bpg4-1* (**d**). Seedlings grown on half-strength MS medium under a growth light (90 μmol photons m$^{-2}$ s$^{-1}$, long-day conditions) for 8 days were transferred to high-light conditions (700 μmol photons m$^{-2}$ s$^{-1}$, continuous light) for 0, 12, or 24 h. The means and SDs were obtained from at least 15 plants and seven biological replicates (**c**), or obtained from approximately 50 mg plant tissue, and eight biological replicates (**d**). Precise values of plant numbers (**c**) or plant weight (**d**) for each data are provided in Source data file. The different letters above the bars indicate statistically significant differences between the samples (two-way ANOVA: Tukey–Kramer's multiple comparisons test, $P < 0.01$), and groups with the same letters are not significantly different. **e** Changes in Fv/Fm during the high-light treatment for up to 24 h. WT, *bpg4-1*, and *BPG4-OX-2* seedlings grown on half-strength MS medium under a growth light (90 μmol photons m$^{-2}$ s$^{-1}$, long-day conditions) for 11 days were transferred to high-light conditions (900 μmol photons m$^{-2}$ s$^{-1}$, continuous light) for 0, 1, 2, 4, 8, 12, or 24 h. Data are presented as the means ± SDs of at least ten biological replicates. Precise values of replicates (*n*) for each data are provided in Source data file.

FACTOR-BINDING PROTEIN 1 (SIB1) and LESION-SIMULATING DISEASE 1 (LSD1) antagonistically regulate GLK transcriptional activity to fine-tune the expression of *PhANGs*, thereby modulating ROS generation[48,56]. BIN2, a BR-signaling kinase, binds strongly to the DBD of GLK1 and weakly to the GCT box, and phosphorylates four Thr sites in the DBD and PRD, which leads to increased stability, DNA-binding and transcriptional activity of GLK1. BIN2 and BPG4 work in BR signaling, but their molecular function on GLK1 appears to be contradictory, suggesting the possibility that BPG4 and BIN2 antagonistically regulate GLK transcriptional activity, similar to SIB1 and LSD1. Future analysis of the possible competitive function between BPG4 and BIN2 could clarify more detailed regulatory mechanisms for GLKs and chloroplast development.

*BPG4-OX* exhibited early flowering, while *BPG4* knockout plants conversely exhibited late flowering (Supplementary Fig. 3). Previous studies reported that *glk1glk2* and *GLK1/2-OX* demonstrated early- and late-flowering phenotypes, respectively, similar to *BPG4*-over-expressing and knockout plants[4,13]. In a recent publication, GLKs were reported to repress flowering by directly activating of the expression of *B-BOX DOMAIN PROTEIN 14/15/16* that suppress the activity of CONSTANS (CO) to bind to the *FLOWERING LOCUS T* (*FT*) promoter[61]. These results implied that suppression of GLKs by BPG4 might influence not only chloroplast development but also flowering time by regulating *FT* expression.

BR has been widely recognized for suppressing chloroplast development, and recent studies have suggested that GLKs function as

central factors of chloroplast development in BR signaling. *det2, bri1-5,* and *bin2-1* mutants exhibited dark green phenotypes caused by defective BR signaling, while these mutants in the background of the *glk1glk2* mutants did not exhibit a dark green phenotype and instead exhibited a pale green phenotype[20]. The *GLK* expression level was increased in *bri1* but decreased by BL treatment and the BR signaling master transcription factors BIL1/BZR1 and BES1[19,62]. In this report, our results suggest that *BPG4* expression is suppressed by the direct binding of BES1 to the G-box in the *BPG4* promoter region under BR signaling (Fig. 6a–k). The difference in the BES1-binding affinity for G-box 1 and G-box 2 may be attributed to the variation in the sequence surrounding the G-box. Our recent research suggests that BIL1/BZR1 transcription factor members display the significant selectivity for two nucleobases flanking the G-box or the NN-BR-response elements (BRREs) core[63]. The G-box 2 nucleobases sequence (CC|CACGTG|GG) is one of the optimal pairs for BIL1/ BZR1 binding[63], while The G-box 1 nucleobases sequence (AT|CACGTG|GA) is not (Supplementary Fig. 13a), which could contribute to the disparity in the BES1-binding affinity for G-box 1 and G-box 2.

*BPG4* expression was regulated by not only BR signaling but also light and circadian rhythm (Fig. 7a–e). *BPG4* expression was observed specifically in photosynthetic tissues, induced by light, and altered clearly along with the circadian rhythm (Fig. 7a–e). Prior genomic or transcriptome analysis suggested that HY5 and PIFs were potential candidates to regulate *BPG4* expression[40,64]. Since both HY5 and PIFs are key transcription factors in light signaling, the activation of *BPG4* expression by light is likely to result from direct regulation by HY5 and PIFs. HY5 and PIFs have also been reported to be associated with the circadian rhythm[43,65]. Furthermore, a recent investigation demonstrated that BES1, which was suggested to regulate *BPG4* expression in this research, regulates circadian oscillation. Based on these findings, coordinated regulation by multiple transcription factors, such as HY5, PIFs, and BES1, is expected to contribute to the precise and dynamic variation in *BPG4* expression along with the circadian rhythm.

ROS in plants are generated mainly in chloroplasts when light energy becomes excessive for plants. Excessive light energy absorption triggers ROS generation within PSI/II and finally causes severe damage to plants called photoinhibition[66,67]. Recent studies have suggested that uncontrolled activation of GLKs accelerates ROS generation and leads to photoinhibition and cell death[48,56]. In this study, we detected ROS and measured the photosynthetic activity of *BPG4*-deficient mutants and *BPG4-OX* transformants under excessive high-light conditions. Our results indicate that the lack of BPG4 causes an increase in ROS generation and a reduction in photosynthetic activity (Fv/Fm) under high light, while *BPG4* overexpression might be helpful for avoiding ROS generation and photoinhibition (Fig. 8a–e). We considered that *BPG4* deficiency could not suppress the excessive activation of GLKs, accelerated the surplus formation of the photosynthetic apparatus, caused the excess absorption of light energy, and finally led to increased ROS generation and photoinhibition. These results suggested that BPG4 negatively regulates chloroplast development via GLKs to prevent ROS generation and to maintain adequate photosynthesis in chloroplasts.

Based on many previous studies of BR biosynthesis and BR-signaling mutants, it is evident that BR deficiency promotes chloroplast development and chlorophyll biosynthesis[19,20,23]. Furthermore, light also obviously fosters these processes. Previously, many factors, such as GLKs, GNC/CGA1, HY5, PIFs, BIN2, and BES1, have been identified as key players in regulating chloroplast development downstream of light and BR signaling[3,5,6,19,20]. Suppressors of chloroplast development, such as PIFs and BES1, are activated under dark and BR signaling, whereas activators, such as GLKs and BIN2, are activated under light and BR deficiency[1]. The regulation of mRNA expression/protein accumulation of these factors by light and BR signaling is consistent with the molecular functions of these factors in chloroplast

regulation. In contrast to these factors, we suggest that BPG4 is a regulator involved in light and BR signaling, acting as a chloroplast homeostasis factor. Interestingly, *BPG4* expression was induced by both light and BR deficiency, coinciding with *GLK* activation, but the function of BPG4 was to suppress GLK activity and chloroplast development. Thus, the regulation of *BPG4* expression by light and BR signaling appears to be opposite and contradictory to BPG4 function in chloroplast regulation. This apparent contradiction could lead us to consider the physiological significance of BPG4 as a chloroplast homeostasis factor. As excessive transcription of *PhANGs* by GLKs could lead to ROS production and become toxic for plants[48,56], a suppressive mechanism to inhibit excess GLK activity would be needed. In the presence of BPG4 (WT), when light and BR deficiency activate *GLKs*, *BPG4* expression is also triggered as an inhibitor. For example, *BPG4* expression was induced by light exposure much later than that of *GLK* (Fig. 7d), and this delayed induction of *BPG4* expression implies a possible BPG4 role in putting the brakes on unnecessary GLK acceleration. Downstream of GLK, *PhANG* expression was rapidly activated at earlier stage of light exposure, and this acceleration of *PhANG* expression appeared to be braked at later stage, concurrent with the initiation of *BPG4* expression (Fig. 7d). This BPG4-induced suppression of GLK transcriptional activity contributes to fine-tuning *PhANG* expression, thereby maintaining homeostasis in chloroplasts (Fig. 9). In the absence of BPG4 (*BPG4*-knockout plants), light and BR deficiency excessively activates *GLKs*, the expression of *PhANGs*, and thereby the surplus formation of the photosynthetic apparatus. Therefore, ROS generation and photoinhibition by excessive light energy absorption are more likely to occur under environmental stress conditions, such as high light, than in the WT (Fig. 9). In *BPG4*-overexpressing plants, the abundant accumulation of BPG4 protein strongly inhibited GLK transcriptional activity and caused decreased *PhANG* expression, thereby leading to suppression of chloroplast development. Consequently, light energy would be moderately absorbed under excessive high-light conditions, which resulted in reduced ROS generation and protection from photoinhibition (Fig. 9). We considered that the contradiction of *BPG4* mRNA expression and BPG4 protein function in chloroplast development would play an important role in fine-tuning the expression of *PhANGs*, keeping the amounts of chlorophylls and the LHC size suitable, avoiding ROS generation, and maintaining chloroplast homeostasis.

In WT plants under light conditions after having germinated in the dark, *GLKs* and *PhANGs* expression rapidly increased within 2 h and peaked within 12 h, while *BPG4* expression slowly increased in 6 h and continued to increase for 72 h (Fig. 7d, Supplementary Fig. 16). *BPG4* expression appeared to be delayed and to act as though it was chasing the expression of *GLKs* and *PhANGs*. In terms of circadian rhythm, the peak and the bottom of *BPG4* expression also appear to perhaps be chasing those of *GLK2* and *PhANG* expression (Fig. 7e). We considered that the delayed *BPG4* expression following *GLK* and *PhANG* expression might partly contribute to inducing the peak-out of *PhANG* expression, suppressing excessive chloroplast development and chlorophyll biosynthesis, and maintaining homeostasis in chloroplasts.

Recent studies have suggested that optimization of LHC size and the amount of chlorophylls increased photosynthetic efficiency and biomass yield[68–70]. Moreover, *ZmGLK*-overexpressing transformants in rice presented enhanced photosynthesis activity, biomass, and grain yields[71,72]. Tomato *uniform ripening* mutants deficient in *SlGLK2* have been used for producing uniformly ripening fruits since tomato domestication, but *SlGLK2* enhances fruit quality in a light- and auxin-dependent manner[11,73]. Our study on the molecular functions of *BPG4* not only contributes to the understanding of an unknown regulatory mechanism of chloroplast development but also could lead to molecular breeding to generate useful crop plants with increased grain yields or enhanced stress resistance in the future.

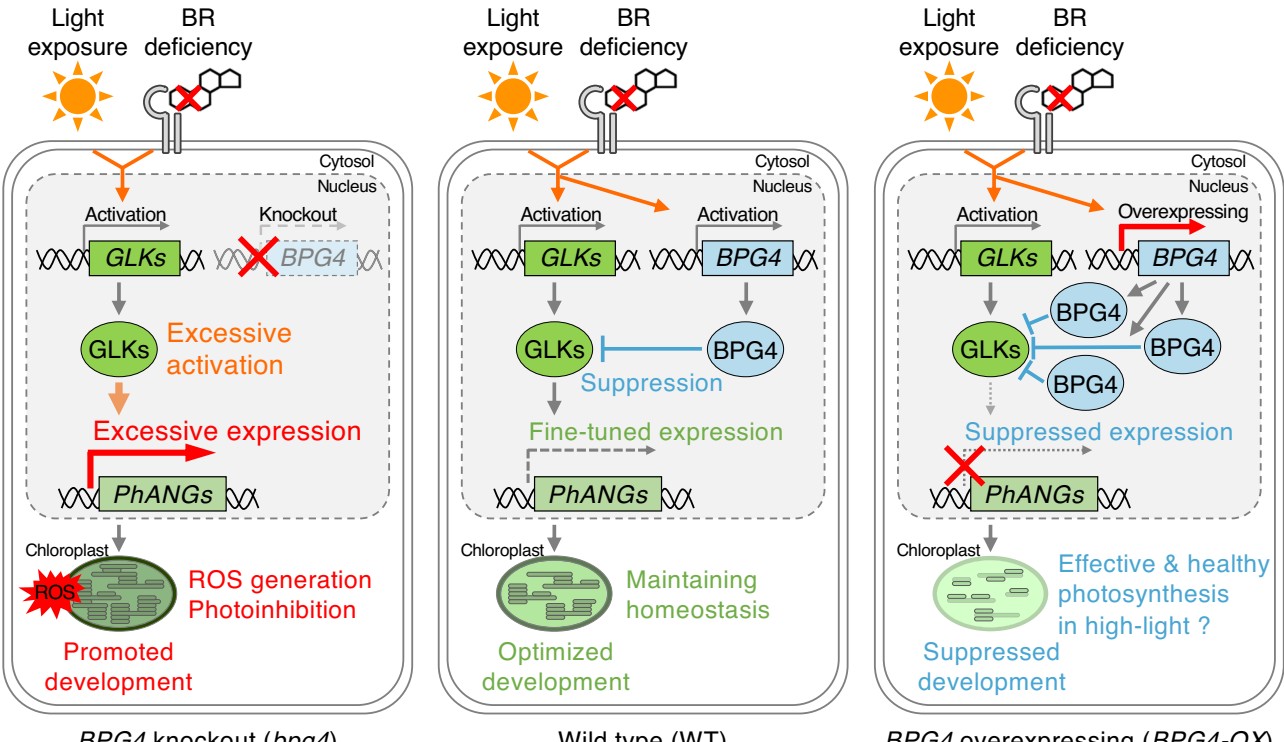

*BPG4* knockout (*bpg4*)   Wild type (WT)   *BPG4* overexpressing (*BPG4-OX*)

**Fig. 9 | Working hypothesis of BPG4 function as a chloroplast homeostasis factor.** Light exposure and BR deficiency induce *GLK* expression, and GLKs promote chloroplast development by enhancing the expression of *PhANGs* such as chlorophyll biosynthesis-related genes and *LHC* genes. Light exposure and BR deficiency also induce *BPG4* expression, but BPG4 suppresses chloroplast development by reducing GLK transcriptional activity. In WT plants, this contradiction between the gene expression and protein function of BPG4 leads to fine-tuned *PhANG* expression, making the amounts of chlorophylls and the size of the LHC antenna suitable, optimizing chloroplast development, and maintaining chloroplast homeostasis (middle panel). In *BPG4*-knockout plants, the lack of *BPG4* leads excessive promotion of GLK transcriptional activity, induces the expression of *PhANGs*, accelerates the surplus formation of the photosynthetic apparatus, causes the excess absorption of light energy, and ultimately triggers ROS generation and photoinhibition (left panel). In *BPG4*-overexpressing plants, the abundant accumulation of BPG4 strongly inhibits GLK transcriptional activity, induces decreased *PhANG* expression, and leads to the suppression of chloroplast development. This sequential regulatory system by BPG4 would contribute to effective and healthy photosynthesis by reducing ROS generation and avoiding photoinhibition under excessive high-light conditions (right panel).

## Methods

### Plant materials and growth conditions

Arabidopsis ecotypes Columbia (Col-0) and Wassilewskija (Ws) were used as WT plants. *bpg4-1* (CS927130), *bpg4-2* (CS391584), *35S:GLK1-GFP/glk1glk2* (CS2107720), and *35S:GLK2-GFP* (CS71752) were obtained from the ABRC. Disruption of *BPG4* was confirmed by qRT–PCR. *det2*, *bri1-5*, *bin2-1*, *bes1-D*, *gsk3 quadruple*, *BRI1-OX* and *glk1glk2* were described previously[9,25,74–78]. Seeds were germinated on medium consisting of half-strength Murashige and Skoog (MS) medium (Duchefa, Haarlem, The Netherlands), 0.8% phytoagar (Duchefa) and 1.5% sucrose and then were transplanted into soil. The plants were grown at 22 °C under white light (16 h light/8 h dark photoperiod for long-day conditions).

### Screening of the *bpg4-1D* mutant

Approximately 10,000 Arabidopsis FOX lines[39] were screened on half-strength MS medium supplemented with 1 μM Brz. After growing for nine days in the light, plants whose cotyledons were less green than those of the WT were identified. The chlorophyll contents of the screened plants were measured, and the full-length cDNA in the mutants was analyzed via PCR with specific primers for the *CaMV 35S* promoter and NOS terminator. Overexpression of *BPG4* was confirmed by qRT–PCR. The primers used are listed in Supplementary Table S1.

### Generation of transgenic plants and mutants

The *BPG4*, *BGH2*, and *BGH3* full-length CDS was amplified from Arabidopsis Col-0 cDNA, cloned into a pENTR/D-TOPO vector (Invitrogen,

Carlsbad, CA), and subsequently cloned in accordance with the Gateway strategy (Invitrogen) into the binary vector pGWB2[79] (http://shimane-u.org/nakagawa/gbv.htm) containing a *CaMV 35S* promoter. To analyze the subcellular localization of the BPG4 protein, the *BPG4* full-length CDS without a stop codon was amplified from Arabidopsis Col-0 cDNA, cloned into a pENTR/D-TOPO vector (Invitrogen), and subsequently cloned in accordance with the Gateway strategy (Invitrogen) into the binary vector pGWB5[79] containing a *CaMV 35S* promoter and *GFP*. To generate a construct in which the *BPG4* promoter including the first exon is fused to the GUS fragment, a 2.1 kb fragment including the promoter region and coding region was amplified from Arabidopsis Col-0 genomic DNA. The amplified products were then cloned into a pENTR/D-TOPO vector (Invitrogen) and subsequently cloned into the binary vector pGWB3 in accordance with the Gateway strategy (Invitrogen). The resulting *CaMV 35S-BPG4*, *CaMV 35S-BPG4-GFP*, and *BPG4pro:GUS* fusion constructs were transformed into Col-0 plants via the floral-dip method[80]. The transgenic plants were screened on half-strength MS medium that included 25 mg/L kanamycin or 25 mg/L hygromycin. Transgene expression was confirmed by qRT–PCR, immunoblotting using an anti-BPG4 antibody, or observations of GFP fluorescence and GUS staining. *BES1pro:BES1-GFP* was obtained from Dr. Ana I Caño-Delgado[81]. *GLK1-OX* was as described previously[16]. The primers used are listed in Supplementary Table S2.

*bgh1-1*, *bgh2-1*, *bgh2-2*, *bgh3-1*, and *bgh3-2* mutants were generated using the CRISPR–Cas9 system. Two guide RNAs for each gene, gRNA1 and gRNA2, were designed using CRISPR direct (https://crispr.dbcls.jp/).

gRNA validity and no off-target effects in *BPG4*, *BGH1*, *BGH2*, and *BGH3* were confirmed by the Guide-it sgRNA In Vitro Transcription and Screening System (Takara, Tokyo, Japan). The oligonucleotides for gRNA1 or gRNA2 were cloned into the AarI site of pKIR1.1[82], and the resulting constructs were transformed into Col-0 plants in the same manner as above. Mutated lines were selected by Sanger sequencing, and null segregants of the CRISP–Cas9 transgene were used as each mutant. The primers used for gRNA are shown in Supplementary Table S3.

### Phenotypic analysis for chlorophyll contents and flowering time

Chlorophyll contents were measured as described previously[35,83]. Flowering time was measured as described previously[84,85], by scoring the time when the main inflorescence shoot had elongated to over 1 cm. The numbers of rosette leaves were counted when the plants opened the first flower. Leaves with lengths over 1 cm were counted as rosette leaves.

### RT–PCR and qRT–PCR

RT–PCR and qRT–PCR was performed as described previously[35,36]. The sequences of the gene-specific primers used are listed in Supplementary Tables S4 and S5.

### Phylogenetic inference

Based on the amino acid sequence of BPG4 (At3g55240) obtained from TAIR (https://www.arabidopsis.org/servlets/TairObject?id=40199&type=locus), *BPG4* homologous genes were searched via BLASTP, in which the NCBI reference protein (refseq_protein) and MpTak1_v5.1[86] from MarpolBase were used. On the basis of the amino acid length and identity, the genes with $e$ value = 1e$^{-16}$ or lower were determined to be BPG4 homologous genes among the candidate genes identified via BLASTP. The amino acid sequences of BPG4 homologous genes were aligned using MAFFT[87], in which "auto" setting was used. From the alignment data produced by MAFFT, a phylogenetic tree was generated by using IQ-TREE[88], and the treefile obtained by IQ-TREE was modified by using iTOL[89]. The results of the multiple sequence alignment were visualized by ESPript 3.0 (https://espript.ibcp.fr/ESPript/ESPript/)[90].

### Transmission electron microscopy

The samples were fixed with 2% paraformaldehyde and 2% glutaraldehyde (GA) in 0.05 M cacodylate buffer (pH 7.4) at 4 °C overnight. After this fixation, the samples were washed 3 times with 0.05 M cacodylate buffer for 30 min each and postfixed with 2% osmium tetroxide (OsO$_4$) in 0.05 M cacodylate buffer at 4 °C for 3 h. The samples were subsequently dehydrated in a gradient of ethanol solutions (50%, 70%, 90%, anhydrous); dehydration in the 50% and 70% solutions was performed for 30 min each at 4 °C, dehydration in the 90% solution was performed for 30 min at room temperature, and there were three changes of anhydrous ethanol for 30 min each at room temperature. After these dehydration steps, the samples were continuously dehydrated in anhydrous ethanol at room temperature overnight. The samples were immersed in propylene oxide (PO) two times for 30 min each and were put into a 50–50 mixture of PO and Quetol-651 resin (Nisshin EM, Tokyo, Japan) for 3 h; then, they were transferred to 100% resin for another 3 h. The samples were subsequently allowed to polymerize at 60 °C for 48 h. Ultrathin (80 nm) sections of the polymerized resins were obtained with an ultramicrotome equipped with a diamond knife (Ultracut UCT; Leica, Vienna, Austria), and the sections were mounted onto copper grids. Afterward, the sections were stained with 2% uranyl acetate at room temperature for 15 min and then washed with distilled water followed by a second staining with lead stain solution (Sigma Aldrich, Tokyo, Japan) at room temperature for 3 min. In terms of observations and imaging, the grids were observed under a transmission electron microscope (JEM-1400Plusi; JEOL, Tokyo, Japan) at an acceleration voltage of 100 kV. Digital images (3296 ×2472 pixels) were collected with a charge-coupled device (CCD) camera (EM-14830RUBY2; JEOL, Tokyo, Japan). Microscopy observations were performed by Tokai Electron Microscopy (Nagoya, Japan).

### Production of recombinant proteins

Production of MBP-BES1 DBD was performed as described previously[33]. The *BPG4* full-length CDS was cloned into a pMAL-c4x vector (NEB, Ipswich, MA, USA) by an In-Fusion HD cloning system (Takara). For recombinant protein production, pMAL-c4x-BPG4 was transformed into *Escherichia coli* BL21 (DE3) pLysS. Expression was induced with 0.4 mM isopropyl-β-D-thiogalactoside (IPTG) at 37 °C for 4 h. MBP-BPG4 was subsequently purified using amylose resin (NEB). The production of GST-GLK1 was performed as described previously, with minor modifications[91]. The *GLK1* full-length CDS was cloned into a pGEX-6P-3 vector by an In-Fusion HD cloning system (Takara). For recombinant protein production, the pGEX-6P-3-GLK1 vector was transformed into *Escherichia coli* Rosetta cells (DE3). Expression was induced with 0.5 mM IPTG at 16 °C overnight. GST-GLK1 was purified using Glutathione Sepharose™ 4B (GE Healthcare, Chicago, IL, USA). GST-GLK1 and MBP-BPG4 were concentrated and desalinated using a Vivaspin 20 Centrifugal Concentrator (Sartorius, Göttingen, Germany). The primers used are listed in Supplementary Table S6.

### Immunoblot analysis and generation of an anti-BPG4 antibody

Plants were ground in liquid nitrogen and extracted with boiling Laemmli buffer [50 mM Tris-HCl (pH 6.8), 100 mM dithiothreitol (DTT), 2% (w/v) Sodium Dodecyl Sulfate (SDS), 0.1% (w/v) bromophenol blue, and 10% (w/v) glycerol]. The proteins were then separated by SDS–PAGE (15% acrylamide gel). For immunoblotting of BPG4 and ACTIN, transfer, blocking, primary antibody reaction, secondary antibody reaction, and detection were performed as described previously[92].

For analysis of BPG4 and BIL1/BZR1-GFP (positive control) phosphorylation status, plants (250 mg) were ground in liquid nitrogen, and extracted by grinding with a mortar and pestle with 0.5 mL of cold λPP extraction buffer [1x NEB Buffer for Protein Metallo Phosphatases (PMP), 2.5 mM MnCl$_2$, 1% (w/w) NP-40 and protease inhibitor cocktail (Roche)]. After passing the extracts through a double-layered Miracloth, aliquots (50 μl) of the flow throughs were incubated with or without 400 units of λPP (NEB, P0753) at 30 °C for 30 min, and boiled at 95 °C with 5× Laemmli buffer for 5 min. Immunoblot analysis was performed as described above.

A rabbit polyclonal antibody against BPG4 was generated using a recombinant MBP-BPG4 fusion protein. The performance of the generated anti-BPG4 antibody was confirmed by the use of Col-0, *bpg4-1*, and *BPG4-OX-2*. Anti-BPG4 antibody was used at a 1:30,000 dilution together with anti-rabbit horseradish peroxidase-conjugated secondary antibody (1:90,000; Promega, Madison, WI, USA). Anti-ACTIN antibody (0869100-CF, MP Biomedical, Santa Ana, USA) was used at a 1:20,000 dilution together with anti-mouse horseradish peroxidase-conjugated secondary antibody (1:60,000; Promega).

### ChIP-qPCR

Arabidopsis chromatin immunoprecipitation (ChIP) assays were performed essentially as described previously[93]. The 15-day-old *BES1-pro:BES1-GFP* and WT seedlings were used for ChIP assays using an anti-GFP antibody (Molecular Probes). Precipitated DNA was analyzed using qPCR by the same protocol as qRT–PCR. We estimated the absolute fraction of DNA recovered from the INPUT (% input DNA) by comparing the reaction threshold cycle of the ChIP sample to a dilution of its own INPUT, and the values in WT (Col-0) were set to 1. The primers used are listed in Supplementary Table S7.

## Protoplast transient expression assays

DNA fragments of the promoter region of *HEMA1*(1.7 kb), *GUN4*(1.5 kb), *CAO* (1.2 kb) and *BPG4* (2.5 kb) were amplified and cloned into a pGLHNew_RLH vector obtained from Dr. Nobutaka Mitsuda by an In-Fusion HD cloning system (Takara). The G-box (CACGTG) sequences in the *BPG4* promoter region were mutated to AAAAAA by inverse PCR using PrimeSTAR Max (Takara). The full-length *BES1*, *VP64*, *BPG4*, and *GLK1* CDSs were cloned into a pENTR/D-TOPO vector (Invitrogen) and subsequently cloned via LR recombination into a binary Gateway expression vector pDEST-35SHSP obtained from Dr. Nobutaka Mitsuda. To generate the *35S:BES1-VP64* effector, DNA fragments of *VP64* were amplified and cloned into the *35S:BES1* effector by an In-Fusion HD cloning system (Takara). The *35S:GFP* effector was obtained from Dr. Nobutaka Mitsuda. The reporter construct was cotransformed together with the effectors into Col-0 protoplasts for the transcriptional activity assay described previously[94,95]. The ratios of LUC to REN activity were calculated to define the relative promoter activity. To detect the LUC and REN activity, a Pickagene® Dual Sea Pansy Luminescence Kit (Toyo Bnet, Tokyo, Japan) was used. Data were obtained from four replicates, and the primers used are listed in Supplementary Table S8.

## EMSAs

An EMSA for analyzing the binding of the BES1 DBD to the *BPG4* promoter region was performed as previously described, with minor modifications (fluorescence was detected using a ChemiDoc Touch™ Imaging System)[33]. For analysis of the binding of GLK1 to the CAO promoter region, GST-GLK1 was incubated with MBP-BPG4 or MBP-tag in 19 μl of binding buffer [10 mM Tris-HCl (pH 8.0), 50 mM KCl, 10 mM EDTA, 2.5% glycerol, 1 mM DTT, 0.05 μg/μl poly(dI-dC)] at 25 °C for 25 min after incubation on ice for 5 min. The binding reactions were carried out with 1 μl of carboxyfluorescein (FAM)-labeled 30-mer dsDNAs (5 pmol) by incubation on ice for 30 min. The reactions were resolved by electrophoresis through 1% agarose gels with 1× TAE buffer. The GLK1-binding site in the *CAO* promoter region was described previously[20]. The sequences of the DNA fragments are listed in Supplementary Table S9.

## Fluorescence microscopy

The plants that were transformed with the *CaMV 35S-BPG4-GFP* construct were observed via confocal laser scanning microscopy with an LSM700 microscope (Zeiss, Jena, Germany).

## Immunofluorescence staining

The samples were fixed with FAA solution (3.7% formaldehyde (methanol-free), 5% glacial acetic acid, 50% ethanol, pH 7.4). Tissue sections were deparaffined with xylene, and then rehydrated through an ethanol series and PBS. They were blocked with G-Block (Genostaff, Tokyo, Japan) for 10 min at room temperature and rinsed in TBS 3 times for 5 min at room temperature. Then, 2 μg/ml anti-BPG4 rabbit polyclonal antibody or normal rabbit IgG (Dako, Glostrup, Denmark, #X0936) was labeled at 4 °C overnight, and rinsed in TBS-T (TBS with 0.1% Tween20) twice for 5 min at room temperature followed by TBS. Donkey anti-rabbit IgG secondary antibody, Alexa Fluor™ Plus 647 (Invitrogen #A32795, 1:200 dilution) were added for 60 min at room temperature for detection of BPG4, and nuclei were counterstained with DAPI (Dojindo, Kumamoto, Japan). Tissue sections were rinsed in TBS 3 times for 5 min at room temperature and mounted with ProLong Gold (Invitrogen #P36934). Samples were observed with an ECLIPSE Ni microscope (Nikon, Tokyo, Japan). Experiments and observations were performed by Genostaff.

## Y2H analysis

The *BPG4* full-length CDS was cloned into a pENTR/D-TOPO vector (Invitrogen) and subsequently cloned into the binary Gateway expression pDEST32 bait vector (Invitrogen) by LR recombination. The full-length CDSs encoding full-length BIN2, GLK1, GLK2, HY5, and COP1, and the domain fragment of GLK1 were cloned into a pENTR/D-TOPO vector (Invitrogen) and subsequently cloned into a pDEST22 prey vector (Invitrogen). Y2H analysis was performed as described previously[92]. The primers used are listed in Supplementary Table S10.

## BiFC

The *BPG4*, *GLK1*, and *GLK2* full-length CDSs were cloned into a pENTR/D-TOPO vector (Invitrogen) and subsequently cloned into the binary Gateway expression vector pB4nYGW or pB4cYGW by LR recombination. The resulting vectors encoding nYFP-BPG4, cYFP-GLK1, and cYFP-GLK2 were used for a transient expression analysis in Arabidopsis suspension-cultured cells as described previously[96]. The expression of *nYFP* and *cYFP* was confirmed in suspension-cultured cells, and the cells were observed via an LSM700 microscope (Zeiss). The primers used are listed in Supplementary Table S10.

## CoIP

The *GLK1* and *GLK2* full-length CDSs were cloned into a pENTR/D-TOPO vector (Invitrogen) and subsequently cloned into the binary Gateway expression vector pGWB421 by LR recombination. *35S:BPG4-GFP*, *35S:10x Myc-GLK1*, and *35S:10x Myc-GLK2* were then transformed into *Agrobacterium*. Transient expression assays involving *Agrobacterium* in *Nicotiana benthamiana* were performed as previously described[77]. Leaves were harvested for 46 h after infiltration, and a total of 0.5 g of plant material was frozen in liquid nitrogen, extracted by grinding with a mortar and pestle, and suspended in 1 mL of cold extraction buffer [50 mM Tris-HCl (pH 7.5), 150 mM NaCl, 10% glycerol, 0.1% (w/w) NP-40 and protease inhibitor cocktail (Roche, Basel, Switzerland)]. These lysates were centrifuged at 11,000 rpm for 5 min at 4 °C, and the resulting supernatant was further centrifuged at 11,000 rpm for 5 min at 4 °C. A portion of the supernatant was collected as an input, and the remaining supernatant was incubated together with GFP-Trap A (Chromo Tek, Planegg, Germany) for 2 h at 4 °C. The incubated samples were washed three times with extraction buffer, after which the washed beads were extracted via boiling Laemmli buffer [100 mM Tris-HCl (pH 6.8), 200 mM DTT, 4% (w/v) SDS, 0.2% (w/v) bromophenol blue, and 10% (w/v) glycerol] for 5 min. The extracted proteins were detected via immunoblot analysis in which anti-Myc antibody (Sigma–Aldrich) and anti-GFP antibody (Molecular Probes, Eugene, Oregon, USA) were used. CoIP in Arabidopsis was performed in the same way as above, and the extracted proteins were detected via immunoblot analysis in which anti-BPG4 antibody and anti-GFP antibody (Molecular Probes) were used. The primers used are listed in Supplementary Table S10.

## GUS staining

*BPG4pro:GUS* transgenic plants were used for the histochemical detection of GUS expression. The samples were stained at 37 °C overnight in GUS staining solution as described previously[97].

## Analysis of gene expression along with the circadian rhythm

WT plants were grown in soil under 12 h light/12 h dark conditions and then exposed to continuous light for 48 h. After exposure to continuous light, rosette leaves were sampled in continuous light conditions every 4 h. Gene expression was subsequently analyzed via qRT–PCR.

## Histochemical staining for ROS detection

DAB staining was performed as previously described[98,99]. Seven-day-old seedlings were immersed in a DAB solution [5 mg/ml DAB, pH 3.8] for 12 h under darkness and transferred to half-strength MS liquid medium. After high-light (1200 μmol photons m$^{-2}$ s$^{-}$) treatment for 2 h, the stained seedlings were treated with a bleach solution [60% (v/

v) ethanol, 20% (v/v) acetic acid, 20% (v/v) glycerol] at 100 °C for 5 min. The bleached seedlings were transferred to a storage solution [80% (v/v) ethanol, 20% (v/v) glycerol] and observed under a stereo microscope.

NBT staining was performed as previously described[100]. Eight-day-old seedlings were transferred to high-light conditions (800 μmol photons $m^{-2} s^{-1}$) for 12 or 24 h. After high-light treatment, the seedlings were stained with an NBT solution [0.5 mg $mL^{-1}$ NBT, 10 mM potassium phosphate buffer, 10 mM $NaN_3$] under vacuum and incubated in the dark for 2 h. After incubation, the stained seedlings were bleached, stored, and observed in the same manner as DAB staining.

## $O_2^{\cdot-}$ quantification

$O_2^{\cdot-}$ was quantified as previously described[101]. Fresh seedling samples (15–18 seedlings or 50–60 mg) were ground in 200 μl of homogenization buffer consisting of 195 μM phosphate buffer (pH 7.8), 10 μM hydroxylammonium chloride and 100 μM EDTA-$Na_2$ in an ice bath. The homogenates were centrifuged at 5000 × g for 5 min at 4 °C. Then, 100 μl of the supernatant was transferred into fresh tubes, after which 100 μl of 17 mM sulfanilamide (in 30% acetic acid) and 100 μl of 7 mM naphthalene diamine dihydrochloride were added to each tube, which were then incubated for 10 min at 37 °C. Then, 300 μl of ether was added to each tube, after which the contents were fully blended and centrifuged at 5000 × $g$ for 5 min at room temperature. The absorbance of the lower aqueous phase was measured at 540 nm via a spectrophotometer. Calibration curves were established from 0 to 5 μM $NO_2^-$. The $r^2$ between OD540 and [$NO_2^-$] was 0.9998. On the basis of the reaction $2O_2^- + H^+ NH_2OH \rightarrow H_2O_2 + H_2O + NO_2^-$, the concentration of $O_2^-$ was calculated according to the equation [$O_2^-$] = 2[$NO_2^-$] (μM) from the calibration curve.

## Determination of photochemical efficiency

Plants grown under growth light conditions (90 μmol photons $m^{-2} s^{-1}$; 16 h light/8 h dark cycle) for 11 days were transferred to high-light conditions (900 μmol photons $m^{-2} s^{-1}$, continuous light) for 1, 2, 4, 8, 12, or 24. After dark adaptation for 20 min, the Fv/Fm was measured with a FluorCam system (FluorCam 800MF; Photon Systems Instruments) equipped with a CCD camera and an irradiation system according to the instrument manufacturer's instructions.

## Statistics and reproducibility

All figures and statistical analyses were constructed and performed in the R environment. For multiple comparisons between genotypes or conditions, one-way ANOVA was performed, and Tukey–Kramer's test was subsequently used for multiple comparisons. For two-group comparisons between genotypes or conditions, an F test was performed, and two-sided Welch's $t$ test or two-sided Student's $t$ test was subsequently used. Statistical parameters including the exact value of $n$, the definition of center, dispersion, and precision measures (mean ± SD) and statistical significance are reported in the Figures and Figure Legends. No sample-size calculation was performed. We set sample sizes based on our preliminary data and published papers by other researchers. No data were excluded from analysis. Plant materials were randomly selected from a larger pool of plants. Investigators were not blinded to group allocation during data collection and/or analysis because there is no group allocation involved in this study. The EMSAs in Fig. 6i were repeated at least twice independently and yielded similar results.

## Accession numbers

Sequence data for the Arabidopsis genes studied in this article can be found in The Arabidopsis Information Resource (www.arabidopsis. org) under the following accession numbers: *BPG4/RPGE2/PEL* (AT3G55240), *BGH1/RPGE3* (AT3G28990), *BGH2/RPGE1* (AT5G02580), *BGH3/RPGE4* (AT1G10657), *ACT2* (AT3G18780), *GAPDH* (AT1G13440),

*elF4a* (AT3G13920), *HEMA1* (AT1G58290), *GUN4* (AT3G59440), *CHLH* (AT5G13630), *PORB* (AT4G27440), *CAO* (AT1G44446), *CHLG* (AT3G51820), *HEMG1* (AT4G01690), *DVR/BPG1* (AT5G18660), *LHCB1.2* (AT1G29910), *LHCB6* (AT1G15820), *LHCA1* (AT3G54890), *PAO* (AT3G44880), *rbcS* (AT1G67090), *psbA* (ATCG00020), *HY5* (AT5G11260), *COP1* (AT2G32950), *GLK1* (AT2G20570), *GLK2* (AT5G44190), *DET2* (AT2G38050), *BRI1* (AT4G39400), *BIN2* (AT4G18710), *BIL1* (AT2G30980), *BIL2* (AT2G30980), *ATSK13* (AT5G14640), *BIL1/BZR1* (AT1G75080), and *BES1* (AT1G19350).

## Reporting summary

Further information on research design is available in the Nature Portfolio Reporting Summary linked to this article.

## Data availability

All the unprocessed data, gels, and blots were provided in the Source Data file. Source data are provided in this paper. Source data are provided with this paper.

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

## Acknowledgements

We thank Jianming Li and Yuki Kondo for *gsk3 quadruple* mutant seeds; and Ana I Caño-Delgado for *BES1pro:BES1-GFP* seeds. This work was supported in part by grants from the NARO Bio-oriented Technology Research Advancement Institution (BRAIN) to T.N. and T.A.; a grant from CREST, Japan Science and Technology Agency, to T.N. and T.A.; and grants from JSPS KAKENHI (21K19077, 21H02114) to T.N. This work was supported by Cross-ministerial Strategic Innovation Promotion Program (SIP), "Building a sustainable food chain that provides abundant and nutritious food" (funding agency: Bio-oriented Technology Research Advancement Institution) to T.N. This work was supported by JST SPRING, Grant Number JPMJSP2110, and JSPS KAKENHI Grant Number JP23KJ1200 to R.T.

## Author contributions

R.T., S.A., Mo.M., A.Y., and T.N. designed the research. R.T., S.A., Mo.M., A.Y., R.A., T.O., K.N., and J.K. performed the experiments. Mi.M. generated the FOX line. S.N. and T.M. generated the recombinant BES1 DBD proteins. T.I. generated the *GLK1-OX* transgenic plants and helped with the production of recombinant GLK1 proteins. R.T., S.A., and Mo.M. analyzed the data. T.K., M.S., K.I., M.T., and T.A. directed and supervised the project. R.T. wrote the article, with much help from A.Y. and T.N. All the authors reviewed and approved the submitted manuscript.

## Competing interests

The authors declare no competing interests.
