## [Peer Review File · Nature Communications]

BPG4 regulates chloroplast development and homeostasis by suppressing GLK transcription factors and involving light and brassinosteroid signalingREVIEWER COMMENTS

Reviewer #1 (Remarks to the Author):

The manuscript by Tachibana et al reports an interesting story of a small Arabidopsis protein (named BPG4) of 95 amino acid. The phenotypes of overexpressing this gene were previously reported in the 2006 Plant J paper that described the FOX approach, resulting in a gene name of PEL (Pseudo-Etiolation in Light). Several published studies suggested that At3g55240 is a downstream target gene of HY5/PIF1 that were known to be important for light signaling. Gene cluster analyses at ATTED (<https://atted.jp>) revealed that BPG4 is coexpressed with genes of light harvesting complexes and photosystems. However, none of these results shed any light on the potential mechanism by which BPG4 influence chloroplast biogenesis or photosynthesis. Thus, the significance of this study is to reveal a potential mechanism by which BPG4 negatively influence chloroplast development by showing that BPG4 is a nuclear localized protein that directly binds and inhibits GLK1/2, two well-known transcription factors crucial for chloroplast biogenesis.

I have two major concerns with the current manuscript.

1). While the study provided good evidence that BR inhibits the expression of BPG4 via a direct protein (BES1)-DNA (BPG4 promoter) interaction, the physiological significance of BPG4 in BR-mediated chloroplast development remains unclear. BR is known to inhibit the chloroplast development as clearly stated by the authors (see lines of 8-12 on page 13). Previous studies suggested two potential mechanisms for such an inhibition. One involves the BES1-GLK1/2 mediated transcriptional regulation (Plant J. 65:634-646) while the other involves the phosphorylation-mediated GLK1/2 degradation (Dev Cell 56:301-324). However, the negative effects of BES1 on BPG4 and BPG4 on GLK1/2 are inconsistent with the known inhibition of BR on chloroplast development. The three previously published BPG genes are chloroplast-localized proteins that when overexpressed caused a similar pale green phenotype. I am doubtful that the genetic screen (looking for BRZ-insensitive pale-green mutants) performed by the authors would discover important genes that mechanistically link BR signaling to chloroplast development. I suggest that the authors should revise their manuscript by shifting the focusing of the revised manuscript on light regulation of BPG4, its negative interaction with GLK1/2, and its protective role on high-light-induced photoinhibition.

2). As I said above, the significance of this study is to reveal a potential biochemical mechanism, (the nuclear interaction between BPG4 and GLK1/2) by which BPG4 influences the chloroplast development. However, it is important that the authors perform additional experiments to convincingly show that the endogenous BPG4 is a nuclear protein. Given its small size (95 AAs), a BPG4-GFP fusion protein is not appropriate to show that BPG4 is a nuclear protein. The authors have successfully generated a quite-specific antibody for BPG4, and they should perform immunohistochemistry experiment to demonstrate the nuclear localization of the endogenous BPG4. At the minimum, the authors should use a simple subcellular fraction approach to show that BPG4 can be easily detected in the nucleus. It is also important to demonstrate that the endogenous BPG4 interacts with a MYC-tagged GLK1/2 protein in transgenic Arabidopsis plants. Ideally, one could generate transgenic Arabidopsis lines expressing the constitutive nuclear-localized or the cytosol-retained form of BPG4 to functionally demonstrate the importance of the nuclear-localized BPG4 for its inhibitory effect on GLK1/2 activity.

Additional minors comments:

1). A recent paper (cited by the authors) showed that overexpression of At5g02580 (known previously as RPGE1 for Repressor of Photosynthetic genes1 and named BGH2 for BPG4 homologous gene 2 in the current study and) also resulted in the pale-green phenotype in the light whereas its loss-of-function mutant had no observable phenotypic change, suggesting a functional redundancy between these two proteins. Thus, the authors should investigate whether or not overexpression of two other homologs of BPG4 could cause a similar pale-green phenotype. Double mutant (bpg4 bgh2) or even higher order mutant is necessary to reveal the true physiological functions of this family of 4 small proteins in chloroplast development or high-light-induced photoinhibition.

2). It would be interesting to cross the GLK1-OX transgene into the bpg4 mutants to see whether the transgenic mutants exhibits stronger phenotypes (chlorophyll contents and the thylakoid structure). Overexpression of GLK1 in the absence of its inhibitor should have a stronger impact on chloroplast development and expression levels of PhANGs.

3). It is inappropriate to say that BPG4 is an unknown gene (line 9 on page 6) when two names were previously used to describe this gene: PEL and RPGE2.

3). The anti-BPG4 antibody detected two bands that were disappeared in the two knockout bpg4 mutants (Figure 2c, Figure3b, and Figure 6c). It would be interesting to know the relationship between the two bands. A preprotein with its processed product or a phosphorylated band with its unphosphorylated form?

4). The authors should report (with a figure in supplementary data) the adult phenotype of the BPG4-overexpression lines. The 2006 Plant J publication reported a taller plant with early flowering phenotype for an At3g55240-overexpression line.

5). The data presented in Figure 2g seems to be inconsistent with that in Figure 2h. The authors should use a trye representative picture (the middle picture in Figure 2g) to show the impact of the bpg4 mutation on the thylakoid structure.

Reviewer #2 (Remarks to the Author):

This manuscript reports the role “BPG4 regulates chloroplast development and homeostasis by suppressing GLK transcription factors via brassinosteroid signaling”. The subject is very important, the authors developed a gene, BPG4, might be a chloroplast homeostasis factor to protect plant from the excessive promotion of chloroplast development and chlorophyll biosynthesis. In a whole, the paper cannot be accepted due to lack of novelty.

Other comments:

1. The authors provide evidence suggest that the BPG4 expression was suppressed by BR signaling and by BES1, but the genetic evidence was lacking.
2. BPG4 interacted with GLK transcription factors and suppressed GLK1 transcriptional activity, however, the BPG4-suppressed GLK1 need more evidence.
3. Whether BPG4 is responsive to BR treatment need to be further clarified.
4. BPG4 suppressed ROS generation and maintained photosynthetic activity under high-light conditions. However, the detail molecular need to be further clarified, the result seems to slightly link with your topics.

5. BPG4 expression was induced by the circadian rhythm, the detail molecular need to be further clarified.
6. Whether BR influences the activity of BPG4 need to be further considered.

Reviewer #3 (Remarks to the Author):

The manuscript by Tachibana et al reported the identification of a new chloroplast developmental regulator BPG4 involved in brassinosteroid signaling. They found that BR signaling suppresses BPG4 expression via BES1, which directly binds to the G-box of the BPG4 promoter. BPG4 interacts with GLK to inhibit the latter activity, thereby suppressing the expression of photosynthesis-related genes. The authors also mentioned that BPG4 acted as a homeostasis factor to maintain effective photosynthesis under excessive high-light conditions by reducing ROS generation. This study provided a mechanism of how BR signaling regulates chloroplast development. However, there is not sufficient and convincing evidence to support the conclusion, and some results are not clearly explained.

1. BPG4 has three other homologs, and what are their expression patterns? Considering the redundant function between homologs and common feedback regulation, the expression level of BGH1-3 should be checked in the bpg4 deficient mutants. Whether these higher-order mutants exhibit more obvious phenotype than the bpg4 single mutant?
2. BR signaling suppressed chlorophyll biosynthesis and chloroplast development. BR signaling inhibited BPG4 expression, but BPG4 suppressed chlorophyll biosynthesis and chloroplast development. It is confusing how BPG4 played a opposite function to regulate chloroplast development if it participated in BR signaling. In the discussion, the authors mentioned that BIN2 phosphorylated and activated GLK to induce the expression of photosynthesis genes, and both BPG4 and GLK expression were suppressed by BES1. What's the relationship between the two obvious contradictive manner to regulate the same genes.
3. In fig 2c, 3b and 6c, there are two protein bands to show BPG4. Is the upper band the phosphorylated form? If so, whether BRZ treatment can enhance accumulation of the phosphorylated form. If BPG4 could be phosphorylated by BIN2? Whether the BIN2 inhibitor bikinin can regulate the BPG4 phosphorylation status.
4. It is not clear why HY5, COP1 and GLK were chosen to analyze their interaction with BPG4? A reasonable explanation should be provided.
5. In fig 6 and 7, the authors analyzed BPG4 expression and the physiological significance of BPG4 for plant growth under high-light. However, the authors emphasized the BPG4 function in BR signaling as shown in the title, and did not well explain how BPG4 links light and BR signaling.
6. The introduction is not well written. For example, lacking the information about chloroplast development and the key regulator GLK.

Reviewer #4 (Remarks to the Author):

Through a genetic screen with plant hormone brassinosteroid (BR) biosynthesis inhibitor brassinazole (BRZ), the authors identified BRZ-insensitive-Pale Green 4 (BPG4) gene that regulates chloroplast function via BR signaling. While the gain-of-function BPG4 mutants have pale green phenotype, the loss-of-function BPG4 mutants display a greener phenotype, suggesting that BPG4 is a negative regulator of chloroplast function, which is supported by several lines of experiments. Interestingly, the authors found that BR master

transcription factor BES1 inhibits BPG4 gene expression, likely by binding two G-Box sequences in the BPG4 gene promoter. The authors found that BPG4 interact with GLK1, a transcription factor that mediates chloroplast development and function. BPG4 inhibits GLK1 function by inhibiting its DNA binding activity. Finally, the authors showed that BPG4 is induced by light and regulated by circadian rhythm and that BPG4 suppresses ROS activity and thus maintains chloroplast homeostasis. The study is of potential interests, but several major points need to be addressed:

1. The BR regulation of BPG4 is the most novel finding of the study and needs to be more vigorously established. The BES1 regulation of BPG4 gene promoter needs mutational analysis with G-boxes in the luciferase reporter assay and in vivo evidence using ChIP assay.
2. Continuing on point 1 and maybe more importantly, the authors should test if BPG4 is phosphorylated and stabilized by BIN2 kinase phosphorylation. In Fig 3b, under BRZ treatment (when BIN2 is activated), BPG4 protein accumulates as two forms, and the top one can be BIN2 phosphorylated form. If this is true, the BR regulation of BPG4 can then happen at both BIN2 phosphorylation and BES1 regulation level, which can make the study highly significant. In vitro kinase assay and examination of BPG4 protein level (with BPG4 antibody) in *bri1*, and BIN2 loss-of-function and gain-of-function mutants can help test the hypothesis. It may take some time to finish these experiments, but I think it's worthwhile.
3. The finding that BPG4 interact and inhibit GLK function is of interest, but similar finding is already made with rice homologs (ref 39, although detailed mechanisms are not established there). The authors made more progress in showing that BPG4 inhibits GLK1 DNA binding. How does that happen? Does the BPG4 interact with GLK1 DNA binding domain to block its DNA binding? This can be tested by mapping the interaction domains in each protein.
4. The discussion section should be rewritten to highlight the significance of the new findings, especially after the authors addressed my points 1-3.

REVIEWER COMMENTS

Reviewer #1 (Remarks to the Author):

<Reviewer#1's comment>

The manuscript by Tachibana et al reports an interesting story of a small Arabidopsis protein (named BPG4) of 95 amino acid. The phenotypes of overexpressing this gene were previously reported in the 2006 Plant J paper that described the FOX approach, resulting in a gene name of PEL (Pseudo-Etiolation in Light). Several published studies suggested that At3g55240 is a downstream target gene of HY5/PIF1 that were known to be important for light signaling. Gene cluster analyses at ATTED (<https://atted.jp>) revealed that BPG4 is coexpressed with genes of light harvesting complexes and photosystems. However, none of these results shed any light on the potential mechanism by which BPG4 influence chloroplast biogenesis or photosynthesis. Thus, the significance of this study is to reveal a potential mechanism by which BPG4 negatively influence chloroplast development by showing that BPG4 is a nuclear localized protein that directly binds and inhibits GLK1/2, two well-known transcription factors crucial for chloroplast biogenesis.

<Author's reply>

We appreciate that reviewer #1 read our manuscript carefully with deep insight for previous publication and raised valuable comments that definitely improved the quality of our work. According to reviewer #1's comments, we performed additional experiments and revised our manuscript to establish a clearer focus. Please see the below for details.

<Reviewer#1's comment>

I have two major concerns with the current manuscript.

1). While the study provided good evidence that BR inhibits the expression of BPG4 via a direct protein (BES1)-DNA (BPG4 promoter) interaction, the physiological significance of BPG4 in BR-mediated chloroplast development remains unclear. BR is known to inhibit the

chloroplast development as clearly stated by the authors (see lines of 8-12 on page 13). Previous studies suggested two potential mechanisms for such an inhibition. One involves the BES1-GLK1/2 mediated transcriptional regulation (Plant J. 65:634-646) while the other involves the phosphorylation-mediated GLK1/2 degradation (Dev Cell 56:301-324). However, the negative effects of BES1 on BPG4 and BPG4 on GLK1/2 are inconsistent with the known inhibition of BR on chloroplast development. The three previously published BPG genes are chloroplast-localized proteins that when overexpressed caused a similar pale green phenotype. I am doubtful that the genetic screen (looking for BRZ-insensitive pale-green mutants) performed by the authors would discover important genes that mechanistically link BR signaling to chloroplast development. I suggest that the authors should revise their manuscript by shifting the focusing of the revised manuscript on light regulation of BPG4, its negative interaction with GLK1/2, and its protective role on high-light-induced photoinhibition.

<Author's reply>

Thank you very much for your helpful suggestions.

First, we agree that BPG4 was regulated not only by BR but also by light, and we changed the title of this manuscript to add 'light': 'BPG4 regulates chloroplast development and homeostasis by suppressing GLK transcription factors downstream of light and brassinosteroid signaling.' Furthermore, we incorporated your suggestion to add an explanation for light regulation in the Discussion (p. 25, lines 14–p. 26, lines 2).

Second, in our opinion, it should also be important that BR is known to inhibit chloroplast development, and this working hypothesis would be agreed upon by reviewer #1. At a minimum, *BPG4* mRNA and protein were induced by the BR biosynthesis inhibitor Brz (Figure 6a and b). *BPG4* mRNA was suppressed by BL treatment (Figure 6c), and the BR-signaling positive transcription factor BES1 (Figure 6d–k). These results suggest that BPG4 is regulated downstream of BR.

Third, reviewer #1 wrote, 'The negative effects of BES1 on BPG4 and BPG4 on GLK1/2 are inconsistent with the known inhibition of BR on chloroplast development.' We would like to explain the possible mechanism by which BPG4 regulates chloroplast homeostasis. Brz and light promote plant greening, chloroplast development, *PhANG* expression, and *BPG4* mRNA expression. These results might impress us that BPG4 is a positive regulator of leaf greening

and chloroplast development. Nevertheless, BPG4 directly interacted with GLK1/2 and inhibited the transcriptional activity of GLK1/2 that promote the transcription of chlorophyll biosynthesis enzyme genes. These results might indicate that BPG4 possesses negative activity in leaf greening and chloroplast development. The inconsistency between the transcriptional regulation of *BPG4* mRNA and the molecular function of BPG4 on GLK1/2 would be a difficult but important point in our research. We think that BPG4 plays a role as a homeostasis factor for chloroplast development that suppresses excessive photosynthesis reaction and photoinhibition with excessive ROS production by GLK1/2.

We revised the manuscript and the figure to more easily understand our considerations. Please see our revised manuscript for details in the Results (p. 19, lines 14–p. 20, lines 14), Discussion (p. 24, lines 1–12; p. 26, lines 17– p. 28, lines 4), and the working hypothesis in Figure 9. We hope that the edited section clarifies the significance of our work

<Reviewer#1's comment>

2). As I said above, the significance of this study is to reveal a potential biochemical mechanism, (the nuclear interaction between BPG4 and GLK1/2) by which BPG4 influences the chloroplast development. However, it is important that the authors perform additional experimnts to convincingly show that the endogenous BPG4 is a nuclear protein. Given its small size (95 AAs), a BPG4-GFP fusion protein is not appropriate to show that BPG4 is a nuclear protein. The authors have successfully generated a quite-specific antibody for BPG4, and they should perform immunohistochemistry experiment to demonstrate the nuclear localization of the endogenous BPG4.

<Author's reply>

Thank you for your important and valuable suggestion. We agree with your concern and performed the additional experiments that you suggested. We performed immunofluorescence staining experiments and fluorescence microscopy observations using an anti-BPG4 antibody in *BPG4*-overexpressing plants and the *BPG4*-knockout mutant *bpg4-1*. The identification of nuclei in cells was performed by DAPI, which can stain nuclear DNA. Then, we identified colocalization of the fluorescence signal by anti-BPG4 antibody and DAPI in *BPG4*-overexpressing plants. As a negative control, in the *BPG4*-knockout mutant *bpg4-1*, only a

DAPI signal was observed. (Supplementary Figure 6). Based on these results, we think that BPG4 is actually localized in the nucleus. We have added an explanation to the Results (p. 11, lines 4–8).

<Reviewer#1's comment>

At the minimum, the authors should use a simple subcellular fraction approach to show that BPG4 can be easily detected in the nucleus.

<Author's reply>

As described above, our results of immunofluorescence staining experiments showed that endogenous BPG4 protein was localized in the nucleus. Thus, we think these results are sufficient to prove the nuclear localization of BPG4.

<Reviewer#1's comment>

It is also important to demonstrate that the endogenous BPG4 interacts with a MYC-tagged GLK1/2 protein in transgenic Arabidopsis plants.

<Author's reply>

Thank you for the suggestion. In addition to CoIP analysis between BPG4-GFP and Myc-GLK1/2 in tobacco, we tried to perform a CoIP assay using Arabidopsis plants. We identified the endogenous BPG4 protein signal in the precipitated complex of GFP-tagged GLK1/2 from 35S:GLK1 and 2-GFP transgenic lines obtained from the Arabidopsis Biological Resource Center (ABRC) (Figure 4d, e). We have added Figure 4 d and e and an explanation to the Results (p. 12, lines 18–20).

<Reviewer#1's comment>

Ideally, one could generate transgenic Arabidopsis lines expressing the constitutive unclear-localized or the cytosol-retained form of BPG4 to functionally demonstrate the importance of the nuclear-localized BPG4 for its inhibitory effect on GLK1/2 activity.

<Author's reply>

Thank you for your interesting comments. We detected a major signal of the interaction between BPG4 and GLK1/2 in the nucleus by BiFC analysis (Figure 4b), the inhibition of the DNA-binding activity of the GLK1 transcription factor by BPG4 using EMSA (Figure 5 e and f), and the inhibition of the transcription activity of GLK1 by BPG4 using a transient assay (Figure 5 g and h). Thus, we hypothesized that BPG4 could interact with the GLK1 transcription factor in the nucleus.

To make a 'cytosol-retained form of BPG4' expressed transgenic Arabidopsis might be effective to elucidate the possible function of BPG4 in the nucleus onto the inhibition of GLK1 activity. Nevertheless, BPG4 lacked both a predicted nuclear localization signal (NLS) and a nuclear export signal (NES), and the size of BPG4 appeared to be smaller than the pore size of the nuclear pore complex (NPC). Although we were also interested in this analysis, we could not create transgenic Arabidopsis lines that overexpressed constitutive cytosol-retained or nuclear-localized BPG4.

GLK1 is well known as a transcription factor in the nucleus. In the additional experiment suggested by reviewer #1, we identified colocalization of the fluorescence signal by anti-BPG4 antibody and DAPI in *BPG4*-overexpressing plants. If reviewer #1 completely understood that the possible inhibition of GLK1 by BPG4 would occur in the nucleus, it would be our pleasure.

<Reviewer#1's comment>

Additional minors comments:

1). A recent paper (cited by the authors) showed that overexpression of At5g02580 (known previously as RPGE1 for Repressor of Photosynthetic genes1 and named BGH2 for BPG4 homologous gene 2 in the current study and) also resulted in the pale-green phenotype in the light whereas its loss-of-function mutant had no observable phenotypic change, suggesting a functional redundancy between these two proteins. Thus, the authors should investigate whether or not overexpression of two other homologs of BPG4 could cause a similar pale-green phenotype.

<Author's reply>

Thank you for your detailed and helpful comments. We tried to create *BGH2/3* overexpression

lines and *BGH1/2/3* knockout lines by CRISPR–Cas9 and analyzed the phenotypes of these plants in terms of chlorophyll content. *BGH2-OX* and *BGH3-OX* showed pale green phenotypes similar to *BPG4-OX*, while *BGH1-KO*, *2-KO*, and *3-KO* plants did not exhibit dark green phenotypes, in contrast to the *BPG4* knockout. We added these results in the new Supplementary Figure 5 and explained them in the Results (p. 10, lines 6–14) and Discussion (p. 22, lines 7–p.23, lines 1).

We had much difficulty in obtaining the cDNA sequence because *BGH1* expression could not be detected in most greening tissues, but we finally succeeded in detecting *BGH1* expression only in siliques (Supplementary Figure 4b). As not only the amino acid sequence but also the nucleic acid sequence of *BGH1* is quite close to that of *BPG4*, we could not clone *BGH1* cDNA from silique total RNA and could not generate *BGH1* transgenic lines at this submission. As the tissue-specific expression pattern of *BGH1* completely differed from that of *BPG4*, we considered that there were no redundant functions in greening organs between *BPG4* and *BGH1*.

<Reviewer#1's comment>

Double mutant (*bpg4 bgh2*) or even higher order mutant is necessary to reveal the true physiological functions of this family of 4 small proteins in chloroplast development or high-light-induced photoinhibition.

<Author's reply>

Subsequent to the single mutants, double mutants (*bpg4bgh2* and *bpg4bgh3*) were generated. There were no significant differences at least in greening organs between double mutants and *bpg4-1*. As *BGH2* was mainly expressed in roots and flowers, in contrast to *BPG4*-expressed tissues, *bpg4bgh2* did not show a stronger phenotype in greening tissues than single-knockout plants. Although *BGH3* was expressed in similar tissues as *BPG4*, *bpg4bgh3* also did not show a stronger phenotype in greening tissues than in single-knockout plants. This difference might be a result of the trends in the weaker function of *BGH3* than *BPG4*, but the actual reason is not known at present. Higher-order mutants (*bpg4bgh2bgh3*) are currently being generated. We added these results to Supplementary Figure 5 and explained them in the Results (p. 10, lines

14–20) and Discussion (p. 22. lines 7–p.23, lines 1).

<Reviewer#1's comment>

2). It would be interesting to cross the *GLK1-OX* transgene into the *bpg4* mutants to see whether the transgenic mutants exhibits stronger phenotypes (chlorophyll contents and the thylakoid structure). Overexpression of *GLK1* in the absence of its inhibitor should have a stronger impact on chloroplast development and expression levels of *PhANGs*.

<Author's reply>

You raised an interesting suggestion, and we also expected *bpg4-1GLK1-OX* to exhibit stronger greening phenotypes than *bpg4-1* and *GLK1-OX*. Nevertheless, *bpg4-1GLK1-OX* appeared not to differ from *bpg4-1* and *GLK1-OX*. By detailed analysis, we found that *GLK1* expression in *bpg4-1GLK1-OX* was suppressed to approximately 10% of that in *GLK1-OX* plants. We also investigated the prior generation line and the independently genotyped another line, but the same result was produced.

Some unknown mechanisms might be considered to regulate *35S* promoter-driven *GLK1* mRNA expression/stability expression. Since there was no difference in chlorophyll content or *PhANG* expression, the observation of chloroplasts by TEM was not performed. We would like to know the cause of this interesting phenomenon and analyze it in the future.

We have demonstrated the analysis of chlorophyll contents and the expression of *PhANGs* and *GLK1* (Supplementary Figure 11) and explained this in the Results (p. 14, lines 10–17).

<Reviewer#1's comment>

3). It is inappropriate to say that *BPG4* is an unknown gene (line 9 on page 6) when two names were previously used to describe this gene: *PEL* and *RPGE2*.

<Author's reply>

We have revised the explanation of *BPG4* following your indications (p. 7, lines 1–3).

<Reviewer#1's comment>

3). The anti-BPG4 antibody detected two bands that were disappeared in the two knockout *bpg4* mutants (Figure 2c, Figure3b, and Figure 6c). It would be interesting to know the relationship between the two bands. A preprotein with its processed product or a phosphorylated band with its unphosphorylated form?

<Author's reply>

Thank you for your important comments. Following your comments, we considered and performed additional experiments.

First, we considered that these two bands might not be processed products because there are no recognized splicing variants in BPG4 transcripts.

Second, the possibility of phosphorylation-mediated modification onto the BPG4 two-band signal was analyzed by phosphatase treatment of the BPG4 protein in plant tissues, but the two bands were not affected by the treatment. Additionally, the relationship between the two BPG4 bands and the BR-signaling kinase BIN2, which is reported as a regulator of GLK1/2 by its phosphorylation activity, was analyzed by Y2H and BIN2 inhibitor BIKININ treatment, but we could not identify a relationship between BPG4 and BIN2.

Then, we hypothesized that these two bands may be posttranslationally modified forms but not phosphorylated forms. Elucidating the two BPG4 bands would be interesting, as reviewer #1 suggested, and we will try to clarify the mechanism in the future.

We have added this information (Supplementary Figure 14) and explained it in the Results (p. 16, lines 19–p. 17, lines 9).

<Reviewer#1's comment>

4). The authors should report (with a figure in supplementary data) the adult phenotype of the BPG4-overexpression lines. The 2006 Plant J publication reported a taller plant with early flowering phenotype for an At3g55240-overexpression line.

<Author's reply>

Thank you for providing valuable suggestions. We observed the flowering time and growth speed in *BPG4-OX* and *BPG4-KO* plants. *BPG4-OX* showed early-flowering phenotypes, identical to the results of a previous publication (Ichikawa et al., The Plant journal, 2006).

Moreover, *BPG4* knockout exhibited late-flowering phenotypes, similar to *GLK1-OX* (Water et al. The Plant Journal, 2008). The additional results demonstrated that BPG4 regulated not only chloroplast development but also flowering time. Additional results for the adult phenotype of the *BPG4*-overexpressing and knockout plants are provided in Supplementary Figure 3 and explained in the Results (p. 8, lines 14–18) and Discussion (p. 24, lines 13–20).

<Reviewer#1's comment>

5. The data presented in Figure 2g seems to be inconsistent with that in Figure 2h. The authors should use a trye representative picture (the middle picture in Figure 2g) to show the impact of the *bpg4* mutation on the thylakoid structure.

<Author's reply>

We apologize for the mis-selecting of this image. In accordance with your indication, we have replaced the data with the correct one (Figure 2g).

Reviewer #2 (Remarks to the Author):

<Reviewer#2's comment>

This manuscript reports the role “BPG4 regulates chloroplast development and homeostasis by suppressing GLK transcription factors via brassinosteroid signaling”. The subject is very important, the authors developed a gene, BPG4, might be a chloroplast homeostasis factor to protect plant from the excessive promotion of chloroplast development and chlorophyll biosynthesis. In a whole, the paper cannot be accepted due to lack of novelty.

<Author's reply>

We appreciate you taking the time to offer us your comments and insights related to the paper. According to your comments, we addressed most of the additional experiments you suggested

and revised our manuscript to make it easier to understand. Please see the below for details.

<Reviewer#2's comment>

Other comments:

1. The authors provide evidence suggest that the BPG4 expression was suppressed by BR signaling and by BES1, but the genetic evidence was lacking.

<Author's reply>

Thank you for your suggestion. First, we identified that *BPG4* mRNA expression was lower in *bes1-ID* than in WT (Figure 6d). Additionally, we detected BES1 binding to the BPG4 promoter by ChIP qPCR analysis (new Figure 6f) and EMSA (Figure 6i). Furthermore, BES1 directly regulated BPG4pro:LUC, as detected by transient assay using Arabidopsis WT (new Figure 6h and k). We trust that these five experiments suggest the high possibility of BES1-regulated suppression of *BPG4* expression by direct binding to the *BPG4* promoter by BES1. We explained these experiments in the Results (p. 15, lines 7–p. 16, lines 18).

Nevertheless, we are currently crossing *bes1-D* and *bpg4-1* or *BPG4-OX-2*. However, due to time constraints, we did not obtain the double homozygous line. We would like to analyze them in the near future.

<Reviewer#2's comment>

2. BPG4 interacted with GLK transcription factors and suppressed GLK1 transcriptional activity, however, the BPG4-suppressed GLK1 need more evidence.

<Author's reply>

In accordance with your indication, we have provided two more experiments, HEMA1pro:LUC and GUN4pro:LUC, in the transient assay to add to the CAOpro:LUC data (new Figure 5g, h). Furthermore, we added a CoIP assay between BPG4 and GLK1/2 in Arabidopsis whole plants (new Figure 4d and e) and a Y2H assay using four GLK1 fragments and two BPG4 fragments (new Figure 4f and g). We think these additional experiments provide 'more evidence' for BPG4 directly binding to GLK1 and BPG4-induced inhibition of the transcriptional activity of GLK1 in *PhANG* expression. We also explained these results in the

Results (p. 12, lines 18–p. 13, lines 4; p. 14, lines 1–9) and Discussion (p. 23, lines 2–23).

<Reviewer#2's comment>

3. Whether BPG4 is responsive to BR treatment need to be further clarified.

<Author's reply>

Thank you for raising an important question that improved the quality of our work. Following your comments, we have examined the response of *BPG4* expression to BR treatment. The results suggest not only that 'Brz treatment' induced BPG4 expression but also that 'BL treatment' suppressed *BPG4* expression. We added the new Figure 6c and explained it in the Results (p. 15, lines 4–5).

<Reviewer#2's comment>

4. BPG4 suppressed ROS generation and maintained photosynthetic activity under high-light conditions. However, the detail molecular need to be further clarified, the result seems to slightly link with your topics.

<Author's reply>

Although we do not fully understand the intended meaning of reviewer #2's suggestion of 'molecular', we will try to explain in more detail and clarify our hypothesis that BPG4 suppressed ROS generation. We added a new Figure 8a to further illustrate the association of BPG4 with ROS generation.

In general, excessive photosynthesis reaction causes the production of ROS, and the overproduced ROS induce photoinhibition (p. 19, lines 20–p. 20, lines 5). In our manuscript, we considered that the overactivation of GLK excessively promoted photosynthetic reaction and produced ROS under high-light conditions, and BPG4 negatively regulates GLKs to prevent ROS generation and to maintain adequate photosynthesis in chloroplasts.

Thus, ROS generation in *BPG4-KO* plants was higher than that in WT plants, and ROS generation in *BPG4-OX* plants was lower than that in WT plants (Figure 8a–d). Not only the observation of stained ROS molecules by plant imaging (Figure 8a and b) but also the

quantitative analysis of ROS (Figure 8c and d) supported the hypothesis that BPG4 suppressed ROS production. Furthermore, maintained and higher photosynthetic activity in *BPG4-OX* under high-light conditions was detected (Figure 8e).

We have rewritten the Results (p. 19, lines 14–p. 21, lines 18) and Discussion (p. 26, lines 3–p. 28, lines 4) and revised the working hypothesis in Figure 9.

<Reviewer#2's comment>

5. BPG4 expression was induced by the circadian rhythm, the detail molecular need to be further clarified.

<Author's reply>

In this manuscript, the major difficult but important point of BPG4 physiological significance was the inconsistency between the light- and Brz-inducibility for *BPG4* that would suggest a positive function for chloroplast development, and the suppression of GLK1 activity by BPG4 that would suggest a negative function for chloroplast development. Finally, based on the inconsistency, we tried to argue for the possible function of BPG4 as a homeostasis factor for chloroplast development. Exploring the circadian regulation of *BPG4* expression would be interesting and important, but the analysis of possible BPG4 regulation by circadian rhythm was necessary to investigate many actors in the circadian rhythm. As the first report of BPG4 function, we would like to focus on only light/BR regulation, and we think that it could be enough volume to construct a manuscript. We would like to perform these experiments on the effect of *BPG4* regulation on circadian rhythm in the near future.

In the Discussion section (p. 25, lines 14–p. 26, lines 2), we described our current consideration of the molecular mechanism governing the regulation of *BPG4* expression by the circadian rhythm. It was hypothesized that multiple transcription factors, including BES1, intricately regulate *BPG4* expression.

<Reviewer#2's comment>

6. Whether BR influences the activity of BPG4 need to be further considered.

<Author's reply>

In general, BR is known to inhibit chloroplast development. At a minimum, *BPG4* mRNA and protein were induced by the BR biosynthesis inhibitor Brz (Figure 6a and b). *BPG4* mRNA was suppressed by BL treatment (Figure 6c), and the BR-signaling positive transcription factor BES1 (Figure 6d). These results suggest that *BPG4* mRNA expression was suppressed by BR. Additional experiments on the effect of BR signaling on BPG4 at the protein level were also performed. By using the *35S* promoter-driven *BPG4-OX* line, in which the mRNA expression level was continuously high and was not affected by BL and Brz treatment, BPG4 protein stability was not affected by BL or Brz treatment (Supplementary Figure 15a–d). The additional experiments suggest that BPG4 was regulated by BR signaling at the mRNA level rather than the protein level.

The possible regulation of the BPG4 protein by phosphorylation was analyzed, but we did not identify the effect of phosphorylation on BPG4 protein status (Supplementary Figure 14a).

We added the new Supplementary Figures 14 and 15 and explained them in the Results (p. 16, lines 19–p. 17, lines 16).

Reviewer #3 (Remarks to the Author):

<Reviewer's comment>

The manuscript by Tachibana et al reported the identification of a new chloroplast developmental regulator BPG4 involved in brassinosteroid signaling. They found that BR signaling suppresses BPG4 expression via BES1, which directly binds to the G-box of the BPG4 promoter. BPG4 interacts with GLK to inhibit the latter activity, thereby suppressing the expression of photosynthesis-related genes. The authors also mentioned that BPG4 acted as a homeostasis factor to maintain effective photosynthesis under excessive high-light conditions by reducing ROS generation. This study provided a mechanism of how BR signaling regulates chloroplast development. However, there is not sufficient and convincing evidence to support

the conclusion, and some results are not clearly explained.

<Author's reply>

Thank you for your helpful and constructive comments. The comments have helped us significantly improve the paper. We addressed all of the additional experiments you suggested in your comments and revised our manuscript to make it easier to understand. Please see the below for details.

<Reviewer#3's comment>

1. BPG4 has three other homologs, and what are their expression patterns? Considering the redundant function between homologs and common feedback regulation, the expression level of BGH1-3 should be checked in the *bpg4* deficient mutants.

<Author's reply>

Thank you for your valuable comments. First, the tissue-specific expression levels of BPG4 and three homologous genes were examined by RT-PCR. *BGH2* was mainly expressed in roots and flowers, in contrast to *BPG4*-expressed tissues. *BGH3* was expressed in similar tissues as *BPG4*. We had much difficulty in obtaining the cDNA sequence because *BGH1* expression could not be detected in most green tissues, but we finally succeeded in detecting *BGH1* expression only in siliques (Supplement Figure 4b).

Second, as requested, we have also investigated the possibility for feedback regulation. Interestingly, the expression pattern of the *BPG4* family exhibited considerable diversity, and only *BGH3* expression was regulated in a feedback manner (Supplement Figure 4c).

An explanation was added to the Results (p. 9, lines 15–p. 10, lines 1) and Discussion (p. 22, lines 7–p. 23, lines 1).

<Reviewer#3's comment>

Whether these higher-order mutants exhibit more obvious phenotype than the *bpg4* single mutant ?

<Author's reply>

Thank you for your interesting and important questions. Following your question, double mutants (*bpg4bgh2* and *bpg4bgh3*) were generated. There were no significant differences at least in greening organs between the double mutants and *bpg4-1*.

As *BGH2* was mainly expressed in roots and flowers, in contrast to *BPG4*-expressed tissues, *bpg4bgh2* did not show a stronger phenotype in greening tissues than single-knockout plants. Although *BGH3* was expressed in similar tissues as *BPG4*, *bpg4bgh3* also did not show a stronger phenotype in greening tissues than single-knockout plants. This might be a result of the trends in the weaker function of *BGH3* than *BPG4*, but the actual reason has not yet been revealed.

Higher-order mutants (*bpg4bgh2bgh3*) are currently being generated.

We added these results to Supplementary Figure 5 and explained them in the Results (p. 10, lines 14–20) and Discussion (p. 22, lines 7–p. 23, lines 1).

<Reviewer#3's comment>

2. BR signaling suppressed chlorophyll biosynthesis and chloroplast development. BR signaling inhibited *BPG4* expression, but *BPG4* suppressed chlorophyll biosynthesis and chloroplast development. It is confusing how *BPG4* played a opposite function to regulate chloroplast development if it participated in BR signaling. In the discussion, the authors mentioned that *BIN2* phosphorylated and activated *GLK* to induce the expression of photosynthesis genes, and both *BPG4* and *GLK* expression were suppressed by *BES1*. What's the relationship between the two obvious contradictive manner to regulate the same genes.

<Author's reply>

We appreciate that you raised a quite important concern. As you mentioned, the function of *BPG4* and the regulation of *BPG4* expression by BR signaling appeared to be inconsistent. The major difficult but important point of *BPG4* physiological significance was the inconsistency between the light and Brz inducibility of *BPG4* mRNA that would suggest a positive function for chloroplast development, and the suppression of *GLK1* activity by *BPG4* that would suggest a negative function for chloroplast development. Finally, based on the inconsistency,

we tried to discuss the possible function of BPG4 as a homeostasis factor for chloroplast development. We considered that the overactivation of GLK excessively promoted photosynthetic reaction and produced ROS under high-light conditions, and BPG4 negatively regulates GLKs to prevent ROS generation and to maintain adequate photosynthesis in chloroplasts.

Regarding light signaling, *GLK* expression quickly responded to light treatment, but *BPG4* expression under light exposure was slightly slower than *GLK* expression (Figure 7d). *BPG4* that is expressed in a delayed manner relative to *GLK* suppresses excessive photosynthesis reaction by *GLK1/2* (p. 28, lines 5–14).

Regarding BR signaling, *BPG4* expression was suppressed by BES1 and induced by Brz (Figure 6 a-d). *GLK* expression was also suppressed by BES1 according to previous research (Yu et al. The Plant Journal, 2011). We considered that *BPG4* mRNA expression was activated by BR deficiency, concurrent with the activation of GLKs by BR deficiency, and that BPG4 acts as GLK inhibitor to prevent the excessive activation of GLKs under BR deficiency. BPG4 would function as a chloroplast homeostasis factor to suppress the excessive promotion of GLK-induced photosynthetic reaction.

As reviewer #3 suggested in our discussion in the former version, BPG4 was an inhibitor of GLK1, but BIN2 was an activator of GLK1. Further fine-tuned regulatory mechanisms by the competitive function of BPG4 on BIN2 for GLK1 activity will reveal the unknown mechanism of chloroplast homeostasis control in the future. Please see the detail in the Discussion (p. 24, lines 1–12).

We revised the manuscript and the figure to more easily understand our considerations. Please see our revised manuscript for details in the Results (p. 19, lines 14–p. 20, lines 14), Discussion (p. 24, lines 1–12; p. 26, lines 17– p. 28, lines 4), and the working hypothesis in Figure 9. We hope that the edited section clarifies the significance of our work.

<Reviewer#3's comment>

3. In fig 2c, 3b and 6c, there are two protein bands to show BPG4. Is the upper band the phosphorylated form? If so, whether BRZ treatment can enhance accumulation of the phosphorylated form. If BPG4 could be phosphorylated by BIN2? Whether the BIN2 inhibitor bikinin can regulat the BPG4 phosphorylation status.

<Author's reply>

First, we considered that these two bands might not be processed products because there are no recognized splicing variants in BPG4 transcripts.

Second, the possibility of phosphorylation-mediated modification of the BPG4 two-band signal was analyzed by phosphatase treatment of the BPG4 protein in plant tissues, but the two bands were not affected by general phosphatase treatment (Supplement Figure 14a).

Additionally, the relationship between the two BPG4 bands and the BR-signaling kinase BIN2, which is reported as a regulator of GLK1/2 by its phosphorylation activity, was analyzed by Y2H and the BIN2 inhibitor Bikinin, but we could not identify a relationship between BPG4 and BIN2 (Supplement Figure 14 b and c).

Additional experiments on the effect of BR signaling on BPG4 at the protein level were also performed. By using the *35S* promoter-driven *BPG4-OX* line, in which the mRNA expression level was continuously high and was not obviously affected by BL and Brz treatment, BPG4 protein stability was not affected by BL or Brz treatment (Supplementary Figure 15a–d). The additional experiments suggest that BPG4 was regulated by BR signaling at the mRNA level rather than the protein level.

Then, we hypothesized that these two bands may be posttranslationally modified forms but not phosphorylated forms. Exploring the two bands of BPG4 would be interesting, as reviewer #3 suggested, and we will try to clarify the mechanism in the future.

We have added this information (Supplementary Figure 14 and 15) and explained it in the Results (p. 16, lines 19–p. 17, lines 16).

<Reviewer#3's comment>

4. It is not clear why HY5, COP1 and GLK were chosen to analysis their interaction with BPG4? A reasonable explanation should be provided.

<Author's reply>

We agree that this point requires clarification. To identify the protein with which BPG4 interacted, we first chose the three well-known and well-analyzed nuclear-localized photosynthesis-associated factors HY5, COP1 and GLK for the Y2H assay. Then, we tested

one by one. Fortunately, GLK1/2 were hits. We added an explanation to the Results (p. 11, lines 21–p. 12, lines 2).

<Reviewer#3's comment>

5. In fig 6 and 7, the authors analyzed BPG4 expression and the physiological significance of BPG4 for plant growth under high-light. However, the authors emphasized the BPG4 function in BR signaling as shown in the title, and did not well explain how BPG4 links light and BR signaling.

<Author's reply>

Thank you for your valuable concern. To resolve your concern, we revised this manuscript to add further detailed explanation for the function and physiological significance of BPG4 linked with not only BR signaling but also light signaling. We have revised the title (p. 1, lines 1–3), Abstract (p. 2, lines 1–14), Results (p. 19, lines 14–p. 20, lines 14), Discussion (p. 26, lines 17– p. 28, lines 4), and Figure 9 to be more in line with your comments. Please see the edited sections. We hope that the revised manuscript explains our considerations more clearly.

<Reviewer#3's comment>

6. The introduction is not well written. For example, lacking the information about chloroplast development and the key regulator GLK.

<Author's reply>

We agree with your comment for the Introduction. We have revised the Introduction to be more in line with your comments to add GLK explanations (p. 4, lines 15–p. 5, lines 14). We hope these revisions provide a more informative Introduction.

Reviewer #4 (Remarks to the Author):

<Reviewer#4's comment>

Through a genetic screen with plant hormone brassinosteroid (BR) biosynthesis inhibitor brassinazole (BRZ), the authors identified BRZ-insensitive-Pale Green 4 (BPG4) gene that

regulates chloroplast function via BR signaling. While the gain-of-function BPG4 mutants have pale green phenotype, the loss-of-function BPG4 mutants display a greener phenotype, suggesting that BPG4 is a negative regulator of chloroplast function, which is supported by several lines of experiments. Interestingly, the authors found that BR master transcription factor BES1 inhibits BPG4 gene expression, likely by binding two G-Box sequences in the BPG4 gene promoter. The authors found that BPG4 interact with GLK1, a transcription factor that mediates chloroplast development and function. BPG4 inhibits GLK1 function by inhibiting its DNA binding activity. Finally, the authors showed that BPG4 is induced by light and regulated by circadian rhythm and that BPG4 suppresses ROS activity and thus maintains chloroplast homeostasis. The study is of potential interests, but several major points need to be addressed:

<Author's reply>

Thank you for the thoughtful feedback you provided regarding our manuscript, which has helped us clarify the significance of our work. We agree with you and have incorporated all of your suggestions and revised our manuscript to make it easier to understand. Please see the below for details.

<Reviewer#4's comment>

1. The BR regulation of BPG4 is the most novel finding of the study and needs to be more vigorously established. The BES1 regulation of BPG4 gene promoter needs mutational analysis with G-boxes in the luciferase reporter assay and *in vivo* evidence using ChIP assay.

<Author's reply>

Thank you for suggesting that the experiments improve our work. We tried to perform additional experiments as you suggested. The additional mutational analysis with G-boxes in the luciferase reporter assay indicated that BES1 regulates *BPG4* expression by binding to G-box-2 *in vivo* (Figure 6j, k). One additional ChIP assay suggested that BES1 bound to the BPG4 promoter region *in vivo* (Figure 6f). We also explained these results in the Results (p. 15, lines 16–p. 16, lines 18).

<Reviewer#4's comment>

2. Continuing on point 1 and maybe more importantly, the authors should test if BPG4 is phosphorylated and stabilized by BIN2 kinase phosphorylation. In Fig 3b, under BRZ treatment (when BIN2 is activated), BPG4 protein accumulates as two forms, and the top one can be BIN2 phosphorylated form. If this is true, the BR regulation of BPG4 can then happen at both BIN2 phosphorylation and BES1 regulation level, which can make the study highly significant. In vitro kinase assay and examination of BPG4 protein level (with BPG4 antibody) in *bri1*, and BIN2 loss-of-function and gain-of-function mutants can help test the hypothesis. It may take some time to finish these experiments, but I think it's worthwhile.

<Author's reply>

Thank you for your important comments. Following your comments, additional experiments were performed.

First, we considered that these two bands might not be processed products because there are no recognized splicing variants in the BPG4 transcripts.

Second, the possibility of phosphorylation-mediated modification onto the BPG4 two-band signal was analyzed by phosphatase treatment onto BPG4 protein in plant tissues, but the two bands were not affected by the treatment (Supplement Figure 14 a).

Additionally, the relationship between the two BPG4 bands and the BR-signaling kinase BIN2, which is reported as a regulator of GLK1/2 by its phosphorylation activity, was analyzed by Y2H and the BIN2 inhibitor Bikinin, but we could not identify a relationship between BPG4 and BIN2 (Supplementary Figure 14 b and c). As BPG4 did not interact with BIN2 in yeast and bikinin did not influence the BPG4 protein, we determined that an *in vitro* kinase assay was not necessary.

Additional experiments on the effect of BR signaling on BPG4 at the protein level were also performed. By using *bri1-5*, *BRI1-OX*, *bin2-1*, *gsk3-quadruple*, BPG4 protein stability and status was analyzed. Endogenous BPG4 protein levels were altered in these mutants, but these results would depend on the regulation of *BPG4* mRNA expression by BES1 (Supplementary Figure 15e, f). The additional experiments suggest that BPG4 was regulated by BR signaling at the mRNA level rather than the protein level.

Then, we hypothesized that these two bands may be posttranslationally modified forms but not

phosphorylated forms. Exploring the two bands of BPG4 would be interesting, as reviewer #4 suggested, and we will try to clarify the mechanism in the future.

We have added this information (Supplementary Figure 14 and 15) and explained it in the Results (p. 16, lines 19–p. 17, lines 16).

<Reviewer#4's comment>

3. The finding that BPG4 interact and inhibit GLK function is of interest, but similar finding is already made with rice homologs (ref 39, although detailed mechanisms are not established there). The authors made more progress in showing that BPG4 inhibits GLK1 DNA binding. How does that happen? Does the BPG4 interact with GLK1 DNA binding domain to block its DNA binding? This can be tested by mapping the interaction domains in each protein.

<Author's reply>

We appreciate your interest and agree that the additional experiments could improve our work. Following your suggestion, we have identified the domain required for their interaction in each protein. The A_thal_3526 domain in BPG4 interacted with the GLK C-terminal domain (GCT-box) in GLK1, not the DNA-binding domain (DBD) (Figure 4f, g). We are now investigating the mechanisms in further detail and intend to report them in a later paper.

We explained these results in the Results (p. 12, lines 21–p. 13, lines 4) and Discussion (p. 23, lines 15–23).

<Reviewer#4's comment>

4. The discussion section should be rewritten to highlight the significance of the new findings, especially after the authors addressed my points 1-3.

<Author's reply>

We tried to rewrite and improve the Results (p. 15, lines 16–p. 17, lines 18) and the Discussion (p. 23, lines 15–23) to be more in line with your comments. We hope that the edited section clarifies the significance of our work.

REVIEWERS' COMMENTS

Reviewer #2 (Remarks to the Author):

The authors have sufficiently modified the manuscript to consider reviewer requests. The manuscript is greatly improved and nicely structured. I have no more comments.

Reviewer #4 (Remarks to the Author):

The authors adequately addressed all of my comments. The manuscript should be ready for acceptance. One small point is that the authors should explain why BES1-VP64 (not just BES1) was used in Figure 6 transient gene expression studies.

The comments from Reviewer #4 in regards to authors' response to Reviewer #3:

I believe that the authors adequately addressed reviewer #3's comments. The author's model that BR-deficiency and light activate both GLKs and BPG4; and BPG4 inhibits GLK function to prevent excessive activation of light regulated genes, is reasonable. Many signaling pathways behave like this (for example auxin signaling activates genes for cellular growth as well as AUX/IAA genes that function to repress auxin signaling). I think the authors can use one of their data (Fig 7d) to further support the model: light-induction of GLK1/2 happened much earlier than the light induction of BPG4, consistent with the idea that BPG4 is activated to at a later stage to dampen GLK1/2 functions.

The authors should discuss this point in page 27, after sentence in line 12-13 [In the presence of BPG4 (WT), when light and BR deficiency activate GLKs, BPG4 expression is also triggered as an inhibitor].

In addition, I also suggest the authors to use some professional editing of the manuscript.

<Author's reply to reviewer #4, and editors>

Thank you for your valuable comments. The reason why we used BES1-VP64, not just BES1, in Figure 6k is that we got only vague results using just BES1 but clear and convincing results using BES1-VP64. Actually, we performed transient gene expression analysis using BES1 and the experiments were repeated at least twice independently and yielded similar results (Please refer to below figures). Repeated results in both experiments suggest the following consideration.

- In the *BPG4pro:LUC^{WT}* and *BPG4pro:LUC^{Mu-1}*, the LUC/REN intensity of BES1 was significantly lower than that of Control, suggesting BES1 directly suppressed *BPG4* expression.
- At least when both G-box-1 and -2 were mutated (*BPG4pro:LUC^{Mu-1&2}*), there was no statistically significant difference between Control and BES1, suggesting BES1 directly bound to G-box in the *BPG4* promoter region and suppressed *BPG4* expression.

Certainly, statistical differences between BES1 and Control in *BPG4pro:LUC^{Mu-1&2}* were not detected, but the LUC/REN intensity of BES1 appeared to tend to slightly decrease compared with that of Control. Therefore, we could not get the confident conclusions from these vague results, so we performed the additional experiments using BES1-VP64 and got clear and convincing results in Figure 6k. We are considering that BES1 might cause partly non-specific reduction of the LUC/REN intensity in transient system due to some unknown mechanisms. Actually, in Figure 6k, the LUC/REN intensity of BES1-VP64 decreased compared with Control. Taken together, we could not ignore this possibly non-specific reduction of the LUC/REN intensity caused maybe by BES1, and decided to use BES1-VP64 in our paper. We also think non-specific reduction by BES1 is a part of suppression by BES1, so conclusion from Figure 6h is not altered.

If you offered us to include the data using BES1 or the additional explanation in our manuscript, we are willing to address your request. If you understood our explanation why we used BES1-VP64 and our conclusion that BES1 bound to G-box-2 in *BPG4* promoter to suppressed *BPG4* expression, it would be our pleasure.

1st experiments (n = 4 biological replicates)

2nd experiments (n = 4 biological replicates)